# A feedback queueing network model for traffic signal control at intersections considering congestion propagation in dynamic stochastic environments

Bin Zhao [1,2,3,4☺¤a¤b¤c] *, Yanni Ju[2‡¤d], Shengyang Jiao[5‡¤e], Denghui Yang[5‡]

**1** Sichuan Vocational and Technical College of Communications, Chengdu, Sichuan, China, **2** Intelligent Policing Key Laboratory of Sichuan Province, Sichuan Police College, Luzhou, Sichuan, China, **3** Sichuan Provincial University Key Laboratory of Highway Traffic Safety, Chengdu, Sichuan, China, **4** Engineering Research Center for Road Transport Safety of Sichuan Province, Chengdu, Sichuan, China, **5** School of Automotive and Transportation, Xihua University, Chengdu, Sichuan, China

☺ These authors contributed equally to this work.
‡ YJ, SJ, and DY authors also contributed equally to this work.
¤a Current Address: Sichuan Vocational and Technical College of Communications, Chengdu, Sichuan, China
¤b Current Address: Sichuan Vocational and Technical College of Communications, Chengdu, Sichuan, China
¤c Current Address: Sichuan Vocational and Technical College of Communications, Chengdu, Sichuan, China
¤d Current Address: Intelligent Policing Key Laboratory of Sichuan Province, Sichuan Police College, Luzhou, Sichuan, China
¤e Current Address: School of Automotive and Transportation, Xihua University, Chengdu, Sichuan, China
* dalouluo@svtcc.edu.cn

## Abstract

Capturing congestion propagation among different facilities at intersections in dynamic stochastic traffic environments poses significant challenges, particularly under oversaturated conditions. In this paper, we present an $M_t/G(x)/C/C$ feedback fluid queueing network model to address CPDSE, integrating random traffic demand, time-varying transition probabilities, and state-dependent stochastic service capabilities. A recursive algorithm is developed to analyze the feedback queueing network model. Simulation experiments reveal that the proposed model and algorithm perform effectively, irrespective of variations in traffic intensity. Compared to the mean results of 200 simulations, the average absolute error is 0.5152 vehicles, and the average relative error is 6.43% across three demand scenarios. Based on the proposed feedback queueing network model, two optimization frameworks are established for traffic signal control, aimed at minimizing either the average vehicle delay time or total costs, including fuel consumption. We propose a rolling optimization strategy that incorporates the mesh adaptive direct search algorithm to achieve real-time traffic signal control. Numerical experiments using actual survey data from Kunshan City yield several noteworthy findings: (1) An optimal moderate-sized time step exists for

**Data availability statement:** Data Statement The data utilized in this study primarily consists of foundational model parameters and numerical experiment datasets. (1) For the foundational model parameters, their acquisition methods and value ranges have been comprehensively detailed in the Model Formulation section. (2) The numerical experiment data includes: Traffic demand matrices (Table 4) and Intersection spatial configuration data (Table 5). The methodological workflow in this manuscript is as follows: First, The intersection queueing network model was constructed using spatial configuration data (Table 5). Second,Operational performance metrics were derived by applying traffic demand data (Table 4). Finally, To validate model efficacy, a simulation model was developed using identical datasets (Tables 4 and 5 ) for error analysis. Furthermore, the data used in the Basic optimization model analysis and Extended optimization model analysis are also based on Tables 4 and 5. Conclusion: All datasets referenced in this manuscript are comprehensively presented within the article.

**Funding:** This work was supported by the Intelligent Policing Key Laboratory of Sichuan Province (Grant No. ZNJW2024KFQN010) and the Natural Science Foundation of Sichuan Province (Grant No. 2025NSFSC2000). The recipient of both grants is Bin Zhao.

**Competing interests:** The authors have declared that no competing interests exist.

rolling optimization to minimize either the average delay time or total costs; specifically, an excessively small time step may increase vehicle average delay time or total costs; (2) The percentage of delay reduction achieved by our method, compared to Synchro software, reaches a maximum of approximately 70% when traffic demand is moderate and the initial state is low; and (3) The percentage reduction in average delay or total costs compared to Synchro initially increases and then decreases with rising traffic intensity.

## Introduction

Effective traffic signal control is a crucial strategy for alleviating urban traffic congestion. It manages the flow of vehicles from different directions at signalized intersections by allowing them to pass at staggered intervals, thereby reducing conflicts between vehicles and enhancing overall traffic efficiency. Since the pioneering work [1], research and development in traffic signal control have fallen mainly into three types of control strategies: fixed-time, actuated, and adaptive. The fixed-time control scheme, derived from historical data, cannot adapt to real-time traffic flow dynamics. The actuated strategy collects real-time data from the detectors (such as loop detectors, video detectors, and radars) and develops variable cycle lengths. It is usually used for an isolated intersection. The adaptive strategy utilizes real-time traffic information to predict future traffic conditions. Meanwhile, it optimizes the signal timing parameters (split, offset and cycle length) based on a defined objective function. It is usually applied for trunk roads or road networks. Many adaptive traffic control systems are developed for practical applications, e.g., SCOOT [2], SCATS [3], OPAC [4], PRODYN [5], RHODES [6], and CRONOS [7]. In recent years, some other adaptive strategies are also proposed. Typical works are store-and-forward-based methods (SFM) [8–10], artificial intelligence methods [11–19], simulation-based optimization methods [20–26], and model-based predictive control methods (MPC) [27–31]. In this paper, our work belongs to the MPC method which optimizes the performance over a prediction horizon and obtains the control action [32]. The MPC uses a mathematical model to predict the control signal's effect on traffic congestion (expressed as an objective function). At every sampling time instant of controller, the objective function is reoptimized using the most recent traffic state measurements.

The intersection traffic flow model is the basis of the MPC. Hence, accurately modeling the intersection is of great importance for the development of effective signal control strategies. A vast number of models have been put forward. According to the level of details, these models can be classified into microscopic, mesoscopic, and macroscopic models. Microscopic models could capture the system's details but require extensive calibration work and colossal computation time. Additionally, microscopic models cannot provide a direct input-output relationship. Therefore, they are relatively better suited to offline traffic simulations [33]. As for macroscopic models, traffic dynamics is described by the mainstream deterministic traffic flow theory models, e.g., the kinematic wave model (KWM) [34–36], cell transmission model (CTM)

[37], and METANET model [38]. Macroscopic models are computationally fast and only require simple inputs. However, they lack the capability of adequately modeling the stochastic nature of traffic [39], especially for the real-time MPC signal control, which has been more and more crucial for real-time traffic signal control in order to respond to natural stochastic variations in traffic [40,41]. Therefore, developing a model capable of simultaneously achieving computational efficiency and strong representational capacity holds significant importance for enhancing the effectiveness of adaptive traffic signal control.

In this paper, queuing theory is leveraged in an attempt to achieve the aforementioned objectives. As one of the most popular mesoscopic models, the queuing model has been used to formulate road traffic flow [20–22,42–59]. Compared with macroscopic models, the queuing model can better study stochastic service systems. Although the queueing model is not more accurate than the corresponding discrete-event simulation, it is typically preferred because it may provide exact solutions, allow structural insights, and is faster than the simulation. Therefore, this paper applies the queueing theory to accurately model signalized intersections in dynamic stochastic environments.

However, signalized intersection is a complex dynamic random service system. In addition to the inherent randomness of intersection, the traffic demand and path matrix between facilities are time-varying. Meanwhile, the service ability depends on the system state, i.e., traffic congestion will decrease service ability. Most importantly, it is difficult to capture the Congestion Propagation in Dynamic Stochastic Environments (CPDSE) among different facilities of the intersection. To address the inherent randomness in intersections, some queueing models or shockwave models used the exponential or normal distributions to describe the vehicle arrival-interval and service-time random variables [1,27,60]. To account for the decrease in service ability caused by congestion, the linear or exponential functions were applied in CTM or META-NET models to formulate the state-dependent velocity [61,62]. To describe the traffic dynamics of intersections, most traffic flow models considered the time-varying traffic demand and path matrix. To capture the congestion propagation phenomenon in deterministic traffic environments, CTM models have been developed for signalized intersections [61,63]. The existing mesoscopic traffic flow models consider some items in the system's inherent randomness, state-dependent service ability, time-varying traffic demand and path matrix, and congestion propagation (see the literature review for details), but lack a comprehensive consideration of these characteristics because of the complexity of CPDSE.

In this paper, we identify the fluid queuing network model as a suitable methodology to formulate those aforementioned dynamic, randomness, state dependence, and congestion propagation for signalized intersections with real-time MPC traffic control. To the best of our knowledge, no study simultaneously models these essential characteristics to quickly and accurately formulate the dynamic performance of signalized intersections. We focus on an isolated intersection which is easily generalized to multiple intersections using the corresponding network topology. We divide the intersection into three categories of facilities and model each facility as a queue. A feedback fluid queuing network model is then developed to consider the CPDSE through a succession of facilities where the random traffic demand and path matrix are time-varying and the random service ability is state-dependent.

The detailed contributions of this paper can be summarized as follows:

(1) We develop a $M_t/G(x)/C/C$ fluid queueing network model with a feedback mechanism for signalized intersections to reveal the CPDSE phenomenon. The exponential (M) and general (G) distributions are used to formulate the randomness of vehicle arrival and service, respectively. Here, the general-distribution service time relaxes the strong assumption in the existing studies. The time-varying arrival rate in $M_t$ and path matrix describe the system dynamics. The state-dependent service rate in $G(x)$ depicts the impact of traffic congestion on service. The feedback queue captures the effect of congestion on departure rates of upstream facilities, i.e., congestion propagation. We prove that the feedback queuing network with $N$ facilities can be decomposed into $N$ node feedback queues $M_t/G(x)/C/C$. A recursive algorithm is proposed to solve the feedback queuing network model. The algorithm's computational complexity is a linear function of the number of time slices given the number of facilities. Simulation experiments based

on a real-world instance show that the proposed model and algorithm perform well regardless of variations in traffic intensity. In addition to the aforementioned capability of easily modeling multiple intersections, the proposed network model is also promising to handle incomplete connected automated vehicle environments if two types of vehicles are considered in the network model.

(2) Based on the $M_t/ G(x)/C/C$ feedback queueing network model, a basic optimization model and a rolling optimization strategy are proposed to realize the real-time traffic signal control. The basic model is to minimize vehicles' average delay time in an intersection by considering the CPDSE. The mesh adaptive direct search algorithm is embedded in the rolling optimization strategy to solve the model efficiently and accurately. Then, we extend the model to minimize the total costs of average delay time and average fuel consumption. The optimization objective can also be the maximization of the system throughput of the intersection. These models can formulate the impact of CPDSE on traffic signal control because the feedback queueing network model is called. We implement vast numerical experiments based on a real-world instance with Kunshan city's operation data to demonstrate the optimization models' performance. To understand how CPDSE affects traffic signal control, we compare the models with the famous traffic signal timing software Synchro that does not consider CPDSE. Results show that the basic and extended models significantly reduce the vehicle's average delay time or total costs including fuel consumption in moderate traffic intensity scenarios.

The rest of this paper is organized as follows. Related works are reviewed in Section 2. Section 3 is the preparations for the intersection model, including a problem statement and main assumptions. The feedback queueing network model is developed for the intersection in Section 4. Section 5 formulates the basic and extended optimization models for traffic signal control. Section 6 presents the recursive algorithm and the rolling optimization algorithm for the feedback queuing network model and the traffic signal control model, respectively. The numerical experiments are presented in Section 7. Conclusions and future work are discussed in Section 8.

## Literature review

As mentioned in the introduction, the existing mesoscopic or macroscopic traffic flow models consider some items in time-varying traffic demand, the system's inherent randomness, state-dependent service ability, and congestion propagation.

The most of the existing traffic flow model consider the time-varying traffic demand. For instance, [61,63] used the cell transmission model (CTM), which is proposed by [37], to describe the time-varying traffic flow of signal intersections. [62] integrated the METANET model of [38] and Kashani model of [64] to capture the evolution of dynamic traffic flows in mixed urban and freeway networks. [65] simplified the macroscopic urban traffic model as S* model to describe the dynamic traffic flow and applied the MPC to control and coordinate urban traffic networks. And then, [66] formulate another MILP optimization problem based on the S* model. Some authors used SFM to describe the dynamic traffic flow in MPC. [32] formulated the problem as the quadratic-programming (QP) problem. In this methodology, the dynamic traffic flow process is modeled by the SFM. However, the potentially high online computation burden may limit its further application for real scenarios. In order to reduce the online computing time, many researchers (such as [67–69]) proposed new methods based on Aboudolas's method. Besides, the mixed logical dynamical system, the continuous-time double queue traffic flow model, fluid queue equation, deep Q-Network, the hierarchical distributed model, simplified shockwave model and the dynamic traffic assignment procedure are also used to describe the dynamic traffic flow. [70] formulate an MPC problem as a MILP by the mixed logical dynamical system. [71] The continuous-time double queue traffic flow model is embedded in the MPC framework to describe the time-varying traffic demand and capture queue spillbacks. [72] proposed a distributed MPC to calculate the optimal signal phases. The dynamic traffic flow model described by fluid queue equation. [12] offered a cooperative traffic signal control with traffic flow prediction for a multi-intersection. The dynamic traffic flow predicted by the deep Q-Network. [73] develop the hierarchical distributed model predictive control to improve

hierarchical distributed control performance. [27] proposed a new model that incorporates a simplified shockwave model with traffic diffusion theory, which can reproduce the stochastic nature of traffic dynamics. [29] combined the traffic routing and signal control models in a continuous-time model for simple traffic networks. The dynamic traffic assignment procedure is used to describe the time-varying traffic flow.

The system's inherent randomness is another essential factor in traffic system. The queue theory is suitable to describe the system's inherent randomness. [74] studied the probability distributions of queue lengths at fixed traffic signal. [75] proposed a methodology that combines the Markov queue and the microscopic traffic simulation model to address a signal control problem. [71] model for optimal traffic signal control by embedding the continuous-time double queue traffic flow model. [72] embed the fluid queue equation in the proposed a distributed MPC to calculate the optimal signal phases. [76] proposed a bootstrapping method that combines the reinforcement learning and queueing theory to reduce the computation time.

Congestion propagation occurs when the traffic network is congested. It has a significant impact on the operational performance of the road network. There are a series of traffic flow model can be used to describe the congestion propagation. [61,63] proposed a novel traffic signal control formulation based on the CTM by considering the unsaturated conditions and the gridlock conditions. Results show that in gridlock conditions, traffic signal control formulation considering congestion propagation produced a timing plan that was better than conventional queue management practices. Besides, the SFM is also used to capture the congestion propagation. For instance, [32] formulated the problem as the quadratic-programming (QP) problem to minimize the risk of queue spillback. Based on [32], many researchers (e.g., [67–69]) proposed new methods to reduce the online computing time. [62] consider congestion propagation by integrating the METANET model and the Kashani model. Furthermore, the S* model is also used to capture the congestion propagation [65,66]. Simplified shockwave model [27] and queue theory [49,71,72] are also used to describe the congestion propagation.

Besides the time-varying traffic demand, the system's inherent randomness, and congestion propagation, state-dependent service is another key feature of traffic systems. Service capacity declines sharply once congestion occurs, a phenomenon referred to as state-dependent service. To the best of our knowledge, no research has incorporated state-dependent service into models of signalized intersections.

In this paper, we introduce queueing theory as a model for traffic intersections due to its advantages, including high time efficiency, ease of calibration, and the availability of closed analytical solutions. The queueing model has been widely applied in road traffic flow analysis. For instance, [1] employed computer simulation techniques to evaluate and assist in the development of formulas for determining optimal signal timing, including calculations for split and cycle lengths (the Webster model). The Webster model proposed an approximation formula to calculate vehicle's average delay by assuming random arrivals. The approximation formula is, in fact, a queuing model, where the inter-arrival times follow an exponential distribution M. [77] proposed a time-dependent $D_t/D/1$ point queuing model (the famous Vickery's model) with a deterministic arrival and service process to analyze bottlenecks in road systems. [78] presented a discrete time Markovian arrival process which models platooned arrivals in road traffic. It takes into account the bunching of traffic and correlation between headways. [74] studied the probability distributions of queue lengths at fixed traffic signal. And the probabilities of the queue lengths, at the ends of the effective green, actual red and amber periods, are also obtained. [75] proposed a Markov queue with fixed arrival rates (traffic demand) and service rates (service ability) for interrupted urban traffic. [44,52,79] study road traffic flow by the $M/G(n)/C/C$ feedback queuing network model with fixed arrival rates and state-dependent service rates. [76] proposed a bootstrapping method that reduces the computation time needed for learning-based traffic control methods to reach acceptable performance levels. The proposed method models lanes of the road network as queues, and derives results on the average service time of vehicles at the intersection level in order to estimate the agent's policy. [56] propose a time-varying and state-dependent fluid queuing model $M_t/G(x)/C/C$ for road traffic systems. It enhances the previous $M_t/G(x)/C/C$ model by considering time-varying arrival rates. Simulation results

show that the fluid queuing model is efficient and accurate in analyzing road systems. In this paper, we extend [56] as a $M_t/G(x)/C/C$ queueing network model with a feedback mechanism to quickly and accurately capture the dynamic performance in an intersection. The feedback mechanism is designed to describe the CPDSE on the network. The time-varying path matrix is also considered in addition to the dynamic random demand and the state-dependent random service.

We list the detailed characteristics of some closely related references in Table 1 to show this study's innovations.

## Preparations

This section presents preparatory work for the mathematical models. First, we state the problem. Second, some of the main assumptions concerning the queuing systems are clarified. Third, we briefly introduce how to obtain the arrival interval parameters and service time distributions in practical applications.

### Main notations

The main notations used in this paper are listed in Table 2.

### Problem statement

This paper focuses on traffic signal control at an isolated intersection by accurately modeling the intersection's traffic behavior. We describe the intersection as a queuing network system. The intersection is first divided into three categories

**Table 1. Characteristics comparison of related literature.**

| Publications | Dynamic Arrival | Randomness in Arrival | Randomness in Service | State-dependent Service | Congestion Propagation | Traffic flow model | Number of intersection |
|---|---|---|---|---|---|---|---|
| [63] | √ | | | | √ | CTM | 2 |
| [61] | √ | | | | √ | CTM | 2 |
| [32] | √ | | | | √ | SFM | 16 |
| [67] | √ | | | | √ | SFM | 6 |
| [68] | √ | | | | √ | SFM | 7 |
| [69] | √ | | | | √ | SFM | 6 |
| [62] | √ | | | | √ | METANET model, Kashani model | 5 |
| [65] | √ | | | | √ | S* model | 4 |
| [66] | √ | | | | √ | S* model | 23 |
| [70] | √ | | | | | Mixed logical dynamical system | 2 |
| [71] | √ | √ | | | √ | Double queue traffic flow model | 9 |
| [72] | √ | √ | | | √ | Fluid queue equation | 6 |
| [12] | √ | | | | | Deep Q-Network | 16 |
| [73] | √ | | | | | Hierarchical distributed traffic model | 2 |
| [27] | √ | | √ | | √ | Simplified shockwave | 8 |
| [29] | √ | | | | | Dynamic traffic assignment procedure | 1 |
| [74] | | √ | √ | | | queue theory | 1 |
| [75] | | √ | √ | | √ | Microscopic simulation model; queue theory | 15 |
| [76] | | √ | √ | | | Reinforcement Learning; queue theory | 1 |
| This paper | √ | √ | √ | √ | √ | Fluid queue network with a feedback mechanism | 1 |

**Table 2. The main notations used in this paper.**

| parameters | Variable | Description |
|---|---|---|
| Input parameters | $l$ | The length of the road section(kilometer) |
| | $w$ | The number of lanes in the road section(lane) |
| | $f$ | The jam density of road section(veh/(lane•kilometer))) |
| | $v_0$ | The free velocity of vehicles (kilometer/hour) |
| | $v_a$ | The mean velocity of vehicles at system state $x = a$(kilometer/hour) |
| | $v_b$ | The mean velocity of vehicles at system state $x = b$(kilometer/hour) |
| | $pr_{ij}$ | The probability of path selection from node $i$ to $j$(%) |
| Queue variables | $\lambda(t)$ | The time-varying arrival rate(veh/hour) |
| | $x(0)$ | The initial system state(veh) |
| | $C$ | The queueing capacity of the road section(veh) |
| | $\mu(x)$ | The service rate of the queueing system with system state $x$ (veh/hour) |
| Output parameters | $\mu_h$ | The service rate of the artificial node for the feedback model |
| | $x(t)$ | The ensemble average number of vehicles (system state) at time $t$ (veh) |
| | $ET(t)$ | The average sojourn time for vehicles arriving at the system at any time $t$ |
| | $f_{out}(t)$ | System throughput at time $t$ (veh/hour) |
| | $P_0(t)$ | The probability of zero customers at time $t$ |
| | $P_B(t)$ | The customer blocking probability at time $t$ |

of facilities by their feature, i.e., imported road sections, entrance lanes, and exported road sections. We respectively denote $\mathcal{N}_1$, $\mathcal{N}_2$ and $\mathcal{N}_3$ as their sets. We also denote $\{0\}$ as the outside of the intersection. $\mathcal{N} = \mathcal{N}_1 \cup \mathcal{N}_2 \cup \mathcal{N}_3$ represents the set of all facilities. Fig 1 shows a typical cross signalized intersection where $\mathcal{N}_1 = \{1, 2, \ldots, 4\}$, $\mathcal{N}_2 = \{13, 14, \ldots, 20\}$ and $\mathcal{N}_3 = \{5, 6, \ldots, 8\}$. According to the location of each facility, we also denote sets $\mathcal{N}_E$, $\mathcal{N}_S$, $\mathcal{N}_W$ and $\mathcal{N}_N$, which will be used in the delay estimate. $\mathcal{N}_E$ refers to the facilities on the east side of the intersection, including imported road sections, entrance lanes, and exported road sections. Correspondingly, $\mathcal{N}_S$, $\mathcal{N}_W$ and $\mathcal{N}_N$ are the facilities on the south, west, and north sides. $\mathcal{N} = \mathcal{N}_E \cup \mathcal{N}_S \cup \mathcal{N}_W \cup \mathcal{N}_N$. In Fig 1, $\mathcal{N}_E = \{1, 5, 9, 13, 17\}$, $\mathcal{N}_S = \{2, 6, 10, 14, 18\}$, $\mathcal{N}_W = \{3, 7, 11, 15, 19\}$ and $\mathcal{N}_N = \{4, 8, 12, 16, 20\}$. Each facility will be treated as a queue.

The vehicle arrives at the facility $i \in \mathcal{N}$ with time interval $A_i$ (a random variable). Entering the facility means the beginning of the service. We treat each available position in the facility as a service desk, which means there are $C_i$ parallel-serial servers in the queuing system if we denote $C_i$ as the capacity of the facility. The total service time (i.e., the time it takes for a customer to traverse the length of the facility) is $B_i$ (a random variable). Hence, the facility belongs to the $A_i/B_i/C_i/C_i$ queueing system.

Arrival interval $A_i$ is time-varying. During non-peak hours, traffic demand is deficient, and thus, arrival interval $A_i$ is very large most of the time. However, traffic demand becomes huge during peak hours, and thus, arrival interval $A_i$ is very small most of the time.

Total service time $B_i$ is state-dependent. When there are no vehicles in the facility ($x_i = 0$) before a vehicle enters that facility, the vehicle can travel at a free velocity, and thus, total service time $B$ is short. When the system state $x_i$ ($0 \leq x_i \leq C_i$) increases, the vehicle must pass and avoid the surrounding vehicles while traveling forward because of the limited ground space. This leads to a decrease in the mean velocity, and thus, total service time $B_i$ will become longer. If the system state $x_i$ is equal to $C_i$, the traffic flow can be viewed as stopped.

The vehicle immediately leaving from the upstream facility will repeatedly visit the downstream facility when the downstream facility is fully occupied ($x_i = C_i$). This will lead to an increase in total service time $B_i$ of the upstream facility. That

**Fig 1. A typical cross signalized intersection.**

is, congestion is propagated into the upstream facility due to the feedback queue. Hence, the intersection belongs to the $A_i/B_i/C_i/C_i$ feedback queueing network.

Vehicles move from node $i$ to node $j$ by the transition probability $pr_{ij}, i,j \in \mathcal{N} \cup \{0\}$. Transition probability $pr_{ij}$ is time-varying. It is also determined by the traffic signal stage $k \in \mathcal{K}$ where $\mathcal{K}$ denotes the set of signal stages. According to time-varying transition probabilities, the signalized intersection's queueing network has different topologies corresponding to different signal stages. Assume that the intersection in Fig 1 uses a four-phase signal control scheme, i.e.,

$\mathcal{K} = \{1, 2, \ldots, 4\}$. Then, four signal stages correspond to four types of topologies for the intersection queueing network, as shown in Fig 2 (a)–(d).

**Main assumptions**

Before we go any further, some underlying assumptions are presented as follows:

(1) The system state of the queueing system for each facility $i \in \mathcal{N}$ is assumed to be the ensemble average number of vehicles $x_i(t)$ at time $t$, which values in the continuous state space$[0, C_i]$. This is an approximation since the system state is the actual number of vehicles which values in the discrete state space $\{0, 1, \ldots, C_i\}$.

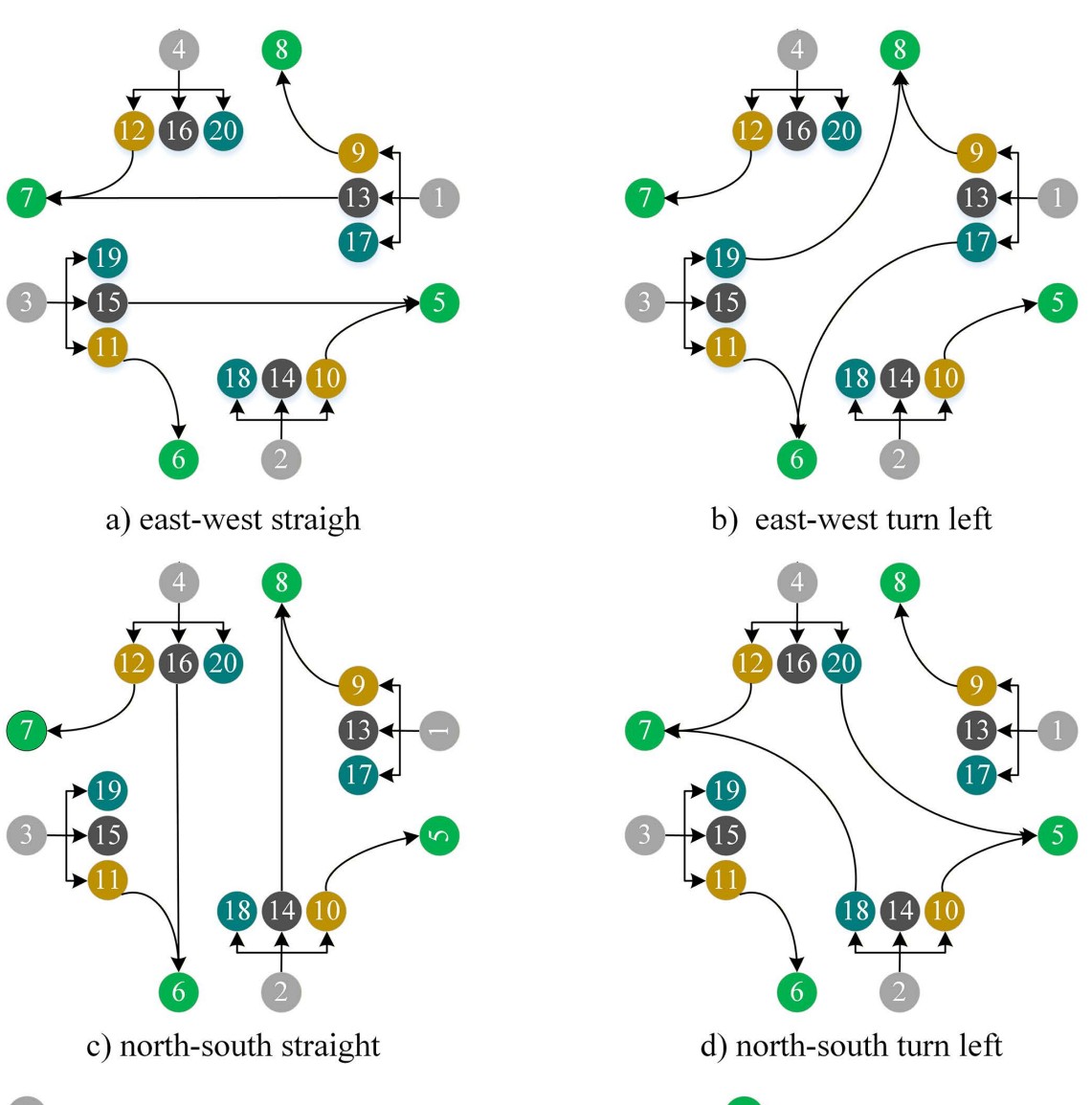

a) east-west straigh

b) east-west turn left

c) north-south straight

d) north-south turn left

① Imported road sections

⑤ Exported road sections

⑨ Right turn lane

⑬ Straight lane

⑰ Left turn lane

**Fig 2. Four signal stages for the intersection.**

(2) Assume that total service time $B_i$ for each facility $i \in \mathcal{N}$ follows a general distribution $G_i(x_i(t))$ with a state-dependent rate $r_i(x_i(t))$.

(3) The external arrival process for each facility $i \in \mathcal{N}$ is assumed as a non-homogeneous Poisson process at a time-varying rate $\lambda_{0i}(t), i \in \mathcal{N}$. The external arrival refers to vehicles arriving at the facility directly from the outside of the intersection. Without loss of generality, we assume arrival rate $\lambda_{0i}(t), i \in \mathcal{N}_2 \cup \mathcal{N}_3$ as zero because vehicles arrive at the entrance lane or the exported road section generally from upstream facilities inside the intersection. That is, only facility $i \in \mathcal{N}_1$ receives vehicles from the outside of the intersection.

(4) Vehicles are assumed to visit facilities repeatedly when facilities suffer from congestion. That is, the intersection is modeled as a feedback queueing network that captures congestion propagation.

## Parameter calibrations

Calibrating the parameters of the intersection queueing network is the first step in practical applications. In this subsection, we briefly introduce the calibration method for the aforementioned essential system parameters, including the facility's capacity, the facility's arrival rate from the external of intersection, the state-dependent service rate for the facility, and the time-varying transition probability between any two facilities.

## Facility capacity

There are two main dimension parameters for each facility: $l_i$ and $w_i$. $l_i$ is the length of a facility in meters, and $w_i$ is the number of lanes in one direction. The capacity of the facility is expressed as

$$C_i = f_i l_i w_i / 1000. \tag{1}$$

where $f_i$ is the jam density, ranging from 115–165 veh/km-lane for road traffic [44].

## External arrival rate

The external arrival rate $\lambda_{0i}(t)$ of the facility $i \in \mathcal{N}$ refers to the number of vehicles per hour arriving at the facility directly from the outside of the intersection. As mentioned in assumptions, no vehicles are arriving at entrance lanes or exported road sections directly from the outside of the intersection, i.e., $\lambda_{0i}(t) = 0, i \in \mathcal{N}_2 \cup \mathcal{N}_3$. Thus, we only need to calibrate $\lambda_{0i}(t), i \in \mathcal{N}_1$. It is easy to obtain and predict the time-varying arrival rate of vehicles $\lambda_{0i}(t), i \in \mathcal{N}_1$ based on the field data collected from imported road sections.

## State-dependent service rate

From Assumption (2), the service rate of each facility $i \in \mathcal{N}$ can be formulated as

$$r_i(x_i(t)) = \frac{v_i(x_i(t))}{l_i}, \tag{2}$$

where $v_i(x_i(t))$ is the mean velocity of vehicles when the system state is $x_i(t)$. State-dependent service rate refers to the phenomenon where the service rate of a road facility varies with the number of vehicles on the road. A prominent example is that when the number of vehicles is low, the road capacity is relatively high, while during congestion, the road capacity significantly decreases. State-dependent service rates have been studied in depth in our previous work [56].The most commonly used mean velocity models, which depend on the discrete system state, were proposed [42]. After that, these models were employed by many research studies in pedestrian traffic and road traffic, such as [43–45,52,79], etc. A linear

model and an exponential model have been proposed [42]. In this paper, we use the exponential model, which is more suitable to the actual situation. The exponential model has a continuous form as follows:

$$v_i(x_i(t)) = v_{i0}\exp(-(\frac{x_i(t)}{\beta_i})^{\gamma_i}),$$

(3)

where

$$\gamma_i = \frac{\ln(\frac{\ln(\frac{v_{ia}}{v_{i0}})}{\ln(\frac{v_{ib}}{v_{i0}})})}{\ln(\frac{a}{b})},$$

$$\beta_i = \frac{a}{(\ln(\frac{v_{i0}}{v_{ia}}))^{\frac{1}{\gamma}}} = \frac{b}{(\ln(\frac{v_{i0}}{v_{ib}}))^{\frac{1}{\gamma_i}}}.$$

The necessary representative points for the continuous model are $(0,v_{i0})$, $(a,v_{ia})$ and $(b,v_{ib})$. Based on intersections data in Chengdu city, we present the value ranges for the three representative points in Table 3.

Fig 3 presents the relationship between the mean velocity, travel time, or traffic volume and system state when the above state-dependent velocity model is applied. In Fig 3a), the velocity $v_i$ gradually decreases with the growing system state $x_i$. When the system state $x_i$ equals the facility capacity $C_i$ and downstream facilities have space available, vehicles move at a minimal velocity. Fig 3b) indicates the mean travel time exponentially increases with the system state. In Fig 3c), the traffic volume $\mu_i$ increases first and then decreases as the system state $x_i$. $u_i^{max}$ is the largest traffic volume, i.e., the traffic capacity of facility. $u_i^{jam}$ is the traffic volume in jam density when vehicles can enter downstream facilities. Note that the facility capacity $C_i$ (the maximum number of vehicles at jam density) is different from the traffic capacity $u_i^{max}$ (the maximum traffic flow per hour). In the real world, when downstream facilities have space available, the mean velocity and traffic volume will not decrease to 0 at jam density. And thus, the exponential model is suitable for the actual situation.

### Time-varying transition probability

Time-varying transition probability $pr_{ij}(t)$ is the probability that vehicles move from node $i$ to node $j$. It may be affected by the traffic signal. Thus, we can formulate $pr_{ij}(t)$ as

$$pr_{ij}(t) = pc_{ij}(t)a_{ij}(t), \forall i, j \in \mathcal{N} \cup \{0\}.$$

(4)

where $pc_{ij}(t)$ is the transition probability of potential traffic demand from node $i$ to node $j$ when there is no traffic signal control at time $t$. It can be estimated or predicted by the historical and real-time data collected from the signalized intersection. $a_{ij}(t)$ is a binary variable. If the path from node $i$ to node $j$ is connected, $a_{ij}(t) = 1$.

**Table 3. Value ranges of three representative points.**

| System states | Value ranges of mean velocity (m/s) |
| --- | --- |
| $x_i(t)=0$ | $v_{i0} \in [50, 60]/3.6$ |
| $x_i(t)=a=0.5C_i$ | $v_{ia} \in [20, 30]/3.6$ |
| $x_i(t)=b=C_i$ | $v_{ib} \in [5, 10]/3.6$ |

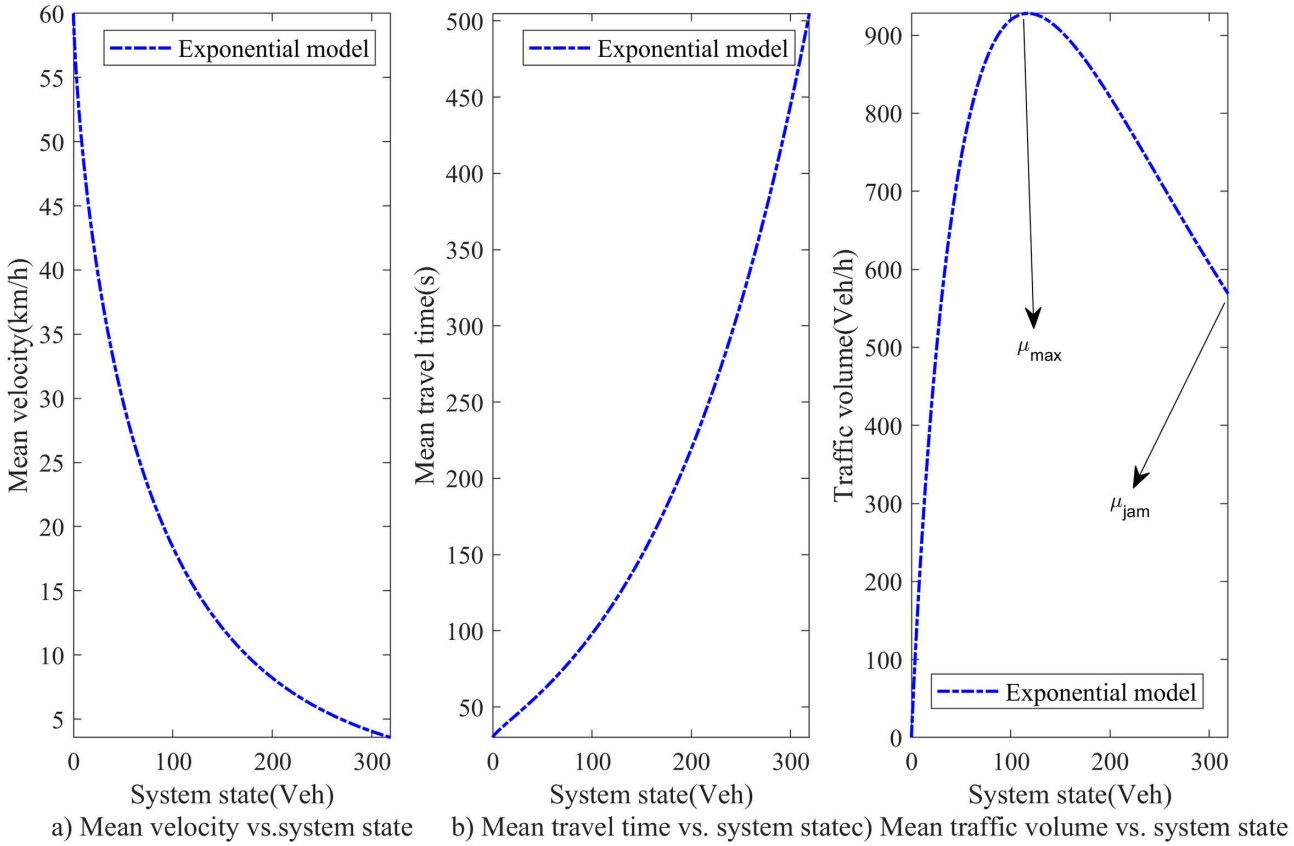

**Fig 3. The relationship between mean velocity, travel time, traffic volume and system state.**

The matrix $(a_{ij}(t))_{(N+1)\times(N+1)}$ defines the network topology at any time where $N$ is the number of facilities. Take the typical cross signalized intersection in Fig 1 as an example, $N = 20$. Fig 4 shows that its traffic signal timing plan where $t_0$ is the initial time, and $g_k, y_k$ and $r_k$ are respectively green-, yellow-, and all-red times at signal phase $k \in \mathcal{K}$. Thus, the network topology defined by $(a_{ij}(t))_{21\times 21}, t \in [t_0, t_0 + g_1]$ is just Fig 2(a). Similarly, Fig 2 (b)-(d) are also defined by $(a_{ij}(t))_{21\times 21}$.

## Queuing Network Model

In this section, we develop a time-varying and state-dependent feedback queuing network model for the signalized intersection. It captures the congestion propagation through a succession of facilities in a dynamic stochastic environment. We first analyze departure processes of facility queues which are essential for the queue decomposition idea of the network model. Then, we extend the generalized expansion methods to formulate our feedback queuing network model. Finally, we obtain the dynamic performance measures based on the proposed queueing network model.

### Departure process of facility queue

Let $\lambda_i(t), i \in \mathcal{N}$ be the time-varying arrival rate for each facility $i \in \mathcal{N}$. From Assumption (3), all vehicles enter facility $i \in \mathcal{N}_1$ (imported road sections) from the outside of the intersection. Thus, the arrival process for facility $i \in \mathcal{N}_1$ is a non-homogeneous Poisson process at the following time-varying rate:

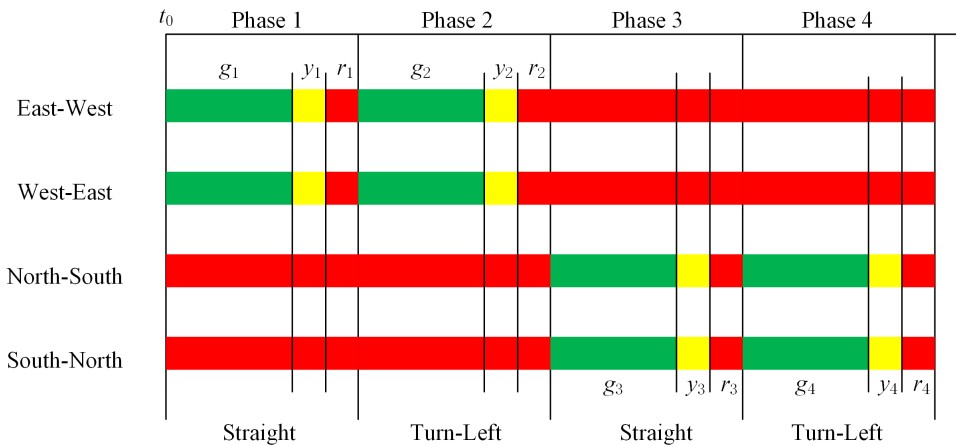

**Fig 4. A traffic signal timing plan for the intersection.**

$$\lambda_i(t) = \lambda_{0i}(t), i \in \mathcal{N}_1,$$

(5)

where $\lambda_{0i}(t)$ is the external arrival rate, which is an input parameter. That is, arrival interval $A_i, i \in \mathcal{N}_1$ follows exponential distribution $M_i(t)$ with time-varying rate $\lambda_i(t)$.

According to Assumption (2), total service time $B_i$ for facility $i \in \mathcal{N}_1$ follows general distribution $G_i(x_i(t))$ with state-dependent rate $r_i(x_i(t))$. Thus, facility $i \in \mathcal{N}_1$ can be modeled as $M_i(t)/G_i(x_i(t))/C_i/C_i$ queue with $C_i$ parallel-serial servers and state-dependent service rate $r_i(x_i(t))$. In the following, we will prove the departure process of the $M_i(t)/G_i(x_i(t))/C_i/C_i, i \in \mathcal{N}_1$ queue is also a non-homogeneous Poisson process with a time-varying rate.

First, two $M_i(t)/G_i(x_i(t))/C_i/C_i$ queuing models are introduced briefly (See [56] for detail). In [56], we have developed $M_i(t)/G_i(x_i(t))/C_i/C_i$ loss and feedback queue models which are equivalent to $M_i(t)/G_i(x_i(t))/1/C_i$ loss and feedback queues with a single virtual server and state-dependent service rate as follows:

$$u_i(x_i(t)) = x_i(t) \times r_i(x_i(t)) = \frac{x_i(t) \times v_i(x_i(t))}{l_i}, i \in \mathcal{N}_1,$$

(6)

where the new service rate $u_i(x_i(t))$ is just the state-dependent service rate of facility $i \in \mathcal{N}_1$. The relationship between the service rate and the system state is illustrated in Fig 2c). This captures the effect of congestion on the service rate.

Fig 5(a) and (b) show the loss and feedback queue systems where $PB_i^L(x_i^L(t))$ and $PB_i^F(x_i^F(t))$ are the vehicle blocking probability for loss/feedback queues, and the indexes "$L$" and "$F$" refer to the loss/feedback queue. In Fig 5(b), node $h_i$ is the artificial node for the facility with feedback to register blocked vehicles, also known as overflows. With probability $1 - PB_i^L(x_i^L(t))$, the circulation facility is not blocked and vehicles proceed to its queue. With probability $PB_i^L(x_i^L(t))$, an arriving vehicle is blocked at the facility with finite space and vehicles will enter the artificial node. When the facility is blocked, the newly arrived vehicle will incur a delay before joining the queue at the facility. After a delay at the artificial node, an overflow vehicle will be blocked with a new probability $PB_i^F(x_i^F(t))$. With probability $1 - PB_i^F(x_i^F(t))$, it will proceed to the capacitated facility from which it was previously rejected. Otherwise, it must retrace its path through the feedback loop into the artificial node again. The artificial node is modeled as an $M_i(t)/G_i/\infty$ fluid queue with service rate $\sigma_i$, so that there will be no waiting to enter this node.

The $M_i(t)/G_i(x_i(t))/C_i/C_i$ loss queue model has the following fluid flow equation

$$\frac{dx_i^L(t)}{dt} = -\theta_i^L(t) + \lambda_i^L(t)\left(1 - PB_i^L(x_i^L(t))\right),$$

(7)

 

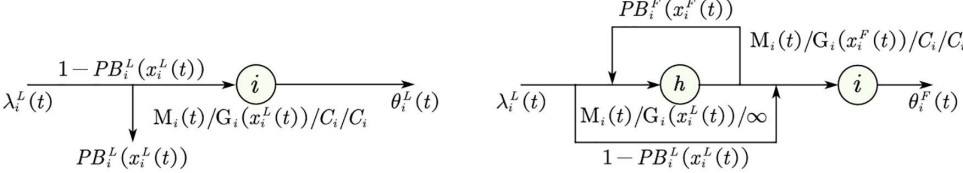

**Fig 5. $M_i(t)/G_i(x_i(t))/C_i/C_i$ loss and feedback queue systems. (a) Loss queue system (b) Feedback queue system.**

$$\theta_i^L(t) = u_i\left(x_i^L(t)\right)\left(1 - PE_i^L\left(x_i^L(t)\right)\right),$$

(8)

where the index "$L$" refers to the loss queue and $x_i^L(t)$ is the corresponding system state. $\theta_i^L(t)$ is the output-flow rate of the loss queue. $u_i(x_i^L(t))$ is just the state-dependent service rate in Equation(8). The probability of zero vehicles $PE_i^L\left(x_i^L(t)\right)$ and the vehicle blocking probability $PB_i^L\left(x_i^L(t)\right)$ have:

$$PE_i^L\left(x_i^L(t)\right) = \exp\left[-\left(\frac{x_i^L(t)}{C_i\beta_{i0}}\right)^{\gamma_{i0}}\right],$$

(9)

$$PB_i^L\left(x_i^L(t)\right) = \exp\left[-\left(\frac{C_i - x_i^L(t)}{C_i\beta_{ic}}\right)^{\gamma_{ic}}\right],$$

(10)

which are obtained by the pointwise stationary fluid flow approximation (PSFFA) method [56]. The PSFFA method performs a pointwise mapping using steady-state queuing relationships. The parameters $\beta_{i0}$ and $\gamma_{i0}$ are calculated by

$$\gamma_{i0} = \frac{\ln\left[\ln PE_{ia}^L / \ln PE_{ib}^L\right]}{\ln(a/b)},$$

(11)

$$\beta_{i0} = \frac{a}{\left[\ln\left(1/PE_{ia}^L\right)\right]^{1/\gamma_{i0}}},$$

(12)

where $PE_{ia}^L$ and $PE_{ib}^L$ are the representative points of $PE_i^L\left(x_i^L(t)\right)$ when the relative density $x_i^L(t)/C_i$ values $a$ and $b$. $PE_{ia}^L$ and $PE_{ib}^L$ are respectively recommended to value 0.81 and 0.21 when $a = 0.1$ and $b = 0.2$.

The parameters $\beta_{ic}$ and $\gamma_{ic}$ are calculated by

$$\gamma_{ic} = \frac{\ln\left(\ln PB_{ia}^L / \ln PB_{ia}^L\right)}{\ln(a/b)},$$

(13)

$$\beta_{ic} = \frac{a}{\left[\ln\left(1/PB_{ia}^L\right)\right]^{1/\gamma_{ic}}}.$$

(14)

where $PB_{ia}^L$ and $PB_{ib}^L$ are the representative points of $PB_i^L\left(x_i^L(t)\right)$ when the relative density $x_i^L(t)/C_i$ values 1-a and 1-b. $PB_{ia}^L$ and $PB_{ib}^L$ are respectively recommended to value 0.83 and 0.44 when $a = 0.1$ and $b = 0.2$.

The $M_i(t)/G_i(x_i(t))/C_i/C_i$ feedback queue model has a more complicated form which is dependent on the $M_i(t)/G_i(x_i(t))/C_i/C_i$ loss queue model:

$$\frac{dx_i^F(t)}{dt} = -\theta_i^F(t) + \lambda_i^L(t)\left(1 - PB_i^L\left(x_i^L(t)\right)\right) + \sigma_i'(t)\rho(t),$$
(15)

$$\theta_i^F(t) = u_i\left(x_i^F(t)\right)\left(1 - PE_i^F\left(x_i^F(t)\right)\right),$$
(16)

where the index "$F$" refers to the feedback queue and $x_i^F(t)$ is the corresponding system state. $\theta_i^F(t)$ is the output-flow rate of the feedback queue. $u_i(x_i^F(t))$ is just the state-dependent service rate in Equation(6). The probability of zero vehicles $PE_i^F\left(x_i^F(t)\right)$ for the feedback queue is obtained by the PSFFA method

$$PE_i^F\left(x_i^F(t)\right) = \exp\left[-\left(\frac{x_i^F(t)}{C_i\beta_{i0}'}\right)^{\gamma_{i0}'}\right].$$
(17)

The parameters $\beta_{i0}'$ and $\gamma_{i0}'$ are calculated by

$$\gamma_{i0}' = \frac{\ln\left[\ln PE_{ia}^F/\ln PE_{ib}^F\right]}{\ln(a/b)},$$
(18)

$$\beta_{i0}' = \frac{a}{\left[\ln\left(1/PE_{ia}^F\right)\right]^{1/\gamma_{i0}'}}.$$
(19)

where $PE_{ia}^F$ and $PE_{ib}^F$ are the representative points of $PE_i^F\left(x_i^F(t)\right)$ when the relative density $x_i^F(t)/C_i$ values $a$ and $b$. $PE_{ia}^F$ and $PE_{ib}^F$ are respectively recommended to value 0.12 and 0.01 when $a = 0.1$ and $b = 0.2$.

The parameter $\sigma_i'(t)$ in Equation (15) is the service rate of new artificial node $h_i'$ after feedback elimination as shown in Fig 6. The feedback arc is removed from the artificial node, and the artificial node is remodeled as an $M_i(t)/G_i/1$ fluid queue with new service rate $\sigma_i'(t)$.

$$\sigma_i'(t) = \left(1 - PB_i^F\left(x_i^F(t)\right)\right)\sigma_i,$$
(20)

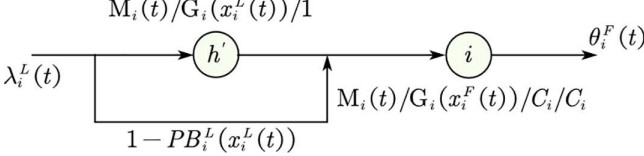

**Fig 6. Feedback elimination for $M_i(t)/G_i(x_i(t))/C_i/C_i$ feedback queue.**

where $\sigma_i$ is the service rate of artificial node $h_i$ with the feedback arc (see Fig 5(b)), i.e., the rate of the delay distribution of a rejected vehicle.

$$\sigma_i = 2\mu_i(C_i)/(1 + cs_i^2), \tag{21}$$

where the service rate in jam density $\mu_i(C_i)$ is obtained by Equation(6). $cs_i$ is the coefficient of variation of service time in jam density. If the service time distribution at the facility is exponential, then, $\sigma_i = \mu_i(C_i)$.

The parameter $PB_i^F(x_i^F(t))$ in Equation (16)is the vehicle blocking probability after a delay at the feedback queueing system's artificial node.

$$PB_i^F(x_i^F(t)) = 1 - \exp\left[-\left(\frac{x_i^F(t)}{C_i\beta_{iC}'}\right)^{\gamma_{iC}'}\right], \tag{22}$$

The parameters $\beta_{iC}'$ and $\gamma_{iC}'$ are calculated by

$$\gamma_{iC}' = \frac{\ln\left[\ln PB_{ia}^F/\ln PB_{ib}^F\right]}{\ln(a/b)}, \tag{23}$$

$$\beta_{iC}' = \frac{a}{\left[\ln\left(1/PB_{ia}^F\right)\right]^{1/\gamma_{iC}'}}. \tag{24}$$

where $PB_{ia}^F$ and $PB_{ib}^F$ are the representative points of $PB_i^F(x_i^F(t))$ when the relative density $x_i^F(t)/C_i$ values $1 - a$ and $1 - b$. $PB_{ia}^F$ and $PB_{ib}^F$ are respectively recommended to value 0.96 and 0.75 when $a = 0.1$ and $b = 0.2$.

The parameter $\rho(t)$ is the utilization rate of a new artificial node $h_i'$

$$\rho(t) = \frac{y_i(t) + 1 - \sqrt{y_i^2(t) + 2cs_i^2 y_i(t) + 1}}{1 - cs_i^2}, \tag{25}$$

where $y_i(t)$ is the system state of a new artificial node $h_i'$. The following fluid flow equation can obtain it

$$\frac{dy_i(t)}{dt} = -\sigma_i'(t)\rho(t) + \lambda_i^L(t)PB_i^L(x_i^L(t)). \tag{26}$$

The following propositions will show the departure process of $M_i(t)/G_i(x_i(t))/C_i/C_i$ loss or feedback queue model is a non-homogeneous Poisson process.

**Proposition 1** *In the* $M_i(t)/G_i(x_i(t))/C_i/C_i, i \in \mathcal{N}_1$ *loss queuing model as described in* Equations (7)-(14), the *departure process (including both vehicles completing service and those that are lost) is a non-homogeneous Poisson process.*

*Proof.* According to [56], the steady-state version of $M_i(t)/G_i(x_i(t))/C_i/C_i$ loss queuing model is just the $M_i/G_i(x_i)/C_i/C_i$ loss queuing model proposed by [43]. Using the steady-state probability distribution $P(x_i)$ of $M_i/G_i(x_i)/C_i/C_i$ loss queuing model, Corollary 1 on page 372 in [43] proves that the departure process of $M_i/G_i(x_i)/C_i/C_i$ loss queuing model is a Poisson process.

According to Equations (7)-(13), the PSFFA method performs a pointwise mapping using steady-state relationships. That is, in the $M_i(t)/G_i(x_i(t))/C_i/C_i$ loss queuing model, any system state $x_i(t)$ is mapped to its probability distribution

$P(x_i(t))$ using the steady-state probability distribution $P(x_i)$ of the $\mathsf{M}_i/\mathsf{G}_i(x_i)/C_i/C_i$ loss queuing model. Thus, the probability distribution $P(x_i(t))$ at any time $t$ has the same form as the steady-state probability distribution $P(x_i)$. Hence, the departure process of $\mathsf{M}_i(t)/\mathsf{G}_i(x_i(t))/C_i/C_i$ loss queuing model is also a Poisson process according to Corollary 1 in Cheah and Smith (1994). As the $\mathsf{M}_i(t)/\mathsf{G}_i(x_i(t))/C_i/C_i$ queue is time-varying, its departure process is a non-homogeneous Poisson process.

**Proposition 2** *In the* $\mathsf{M}_i(t)/\mathsf{G}_i(x_i(t))/C_i/C_i, i \in \mathcal{N}_1$ *feedback queuing model as described in Equations (15)-(22), the departure process (including both vehicles completing service and those that repeatedly visit) is a non-homogeneous Poisson process.*

*Proof.* From Fig 5, the arrival process of artificial node $h_i$ before feedback is a Poisson process with a rate $\lambda_i(t)PB_i^L(t)$. As the artificial node $h_i$ has infinite severs, its departure process is also a Poisson process [80]. Thus, the departure process of vehicles that repeatedly visit the facility is a Poisson process with a time-varying rate, i.e., non-homogeneous Poisson process.

From Fig 5 and Equation(15), the feedback queue's effective arrival consists of the ones from the loss queue and the artificial node. Both of them are Poisson processes. Thus, the effective arrival of the feedback queue is a Poisson process. Hence, the departure process of vehicles completing service, which comes from effective arrival, is a non-homogeneous Poisson process according to Proposition 1.

Propositions 1 and 2 are fundamental properties for the queue decomposition idea of the time-varying and state-dependent feedback queuing network. We use the following proposition to present it.

**Proposition 3** *For the signalized intersection* $\mathsf{A}_i/\mathsf{B}_i/C_i/C_i, \forall i \in \mathcal{N}$ *feedback queuing network with Poisson external arrival processes* $\mathsf{M}_i(t), \forall i \in \mathcal{N}_1$ *and general-distribution service time* $\mathsf{G}_i(x_i(t)), \forall i \in \mathcal{N}$ *as described in Sections 3.1 and 3.2, each facility* $i \in \mathcal{N}$ *is a* $\mathsf{M}_i(t)/\mathsf{G}_i(x_i(t))/C_i/C_i$ *feedback queue.*

*Proof.* As previously mentioned, each facility $i \in \mathcal{N}_1$ is a $\mathsf{M}_i(t)/\mathsf{G}_i(x_i(t))/C_i/C_i$ feedback queue. According to Proposition 2, the departure process of vehicles completing service for facility $i \in \mathcal{N}_1$ is a non-homogeneous Poisson process. Thus, the arrival process for its downstream facility $i \in \mathcal{N}_2$ is also a non-homogeneous Poisson process because of the superposition property of Poisson processes. Then, each facility $i \in \mathcal{N}_2$ is a $\mathsf{M}_i(t)/\mathsf{G}_i(x_i(t))/C_i/C_i$ feedback queue. Similarly, we can prove that each facility $i \in \mathcal{N}_3$ is a $\mathsf{M}_i(t)/\mathsf{G}_i(x_i(t))/C_i/C_i$ feedback queue. Hence, Proposition 3 holds.

Proposition 3 shows that the signalized intersection feedback queuing network with $N$ facilities can be decomposed into $N$ feedback queues $\mathsf{M}_i(t)/\mathsf{G}_i(x_i(t))/C_i/C_i$. Thus, Equations (15)-(19) are still available for each facility queue, which inspire us to develop the following $\mathsf{M}_i(t)/\mathsf{G}_i(x_i(t))/C_i/C_i$ feedback queuing network model for the signalized intersection.

### Feedback queuing network model

From Proposition 3, each facility $i \in \mathcal{N}$ is a $\mathsf{M}_i(t)/\mathsf{G}_i(x_i(t))/C_i/C_i$ feedback queue. Therefore, its fluid flow equation within the network can be rewritten as:

$$\frac{\mathrm{d}x_i^F(t)}{\mathrm{d}t} = -\theta_i^F(t) + \lambda_i^L(t)\left(1 - PB_i^L\left(x_i^L(t)\right)\right) + \sigma_i'(t)\rho(t), \tag{27}$$

where the vehicle blocking probabilities $PB_i^L\left(x_i^L(t)\right)$, the service rate of new artificial node $\sigma_i'(t)$, and the utilization rate of new artificial node $\rho(t)$ are still formulated by Equations (10), (20) and (25) respectively.

$x_i^L(t)$ is the system state when only facility $i$ is modeled as a loss queue (i.e., feedback queues for the other facilities). Its fluid flow equation within the network has the following form

$$\frac{\mathrm{d}x_i^L(t)}{\mathrm{d}t} = -\theta_i^L(t) + \lambda_i^L(t)\left(1 - PB_i^L\left(x_i^L(t)\right)\right), \tag{28}$$

$$\theta_i^L(t) = u_i(x_i^L(t)) \left(1 - PE_i^L\left(x_i^L(t)\right)\right) \sum_{j \in \mathcal{N} \cup \{0\}} pr_{ij}(t), \tag{29}$$

where the service rate $u_i(x_i^L(t))$ and the probability of zero vehicles $PE_i^L\left(x_i^L(t)\right)$ are still formulated by Equations (6) and (9) respectively. The time-varying transition probability $pr_{ij}(t)$, $i, j \in \mathcal{N} \cup \{0\}$ is formulated by Equation (4). Note that $\sum_{j \in \mathcal{N} \cup \{0\}} pr_{ij}(t)$ may be zero when there is no green time for facility $i \in \mathcal{N}_2$ (entrance lane). Take the intersection in Fig 1 as an example. If Fig 4 is its traffic signal timing plan, $\sum_{j \in \mathcal{N} \cup \{0\}} pr_{ij}(t)$ is zero for facility $i = 14, 16, 18, 20$ at time $t \in [t_0, t_0 + g_1]$ as shown in Fig 2(a). Equations (28) and (29) indicate that the fluid flow equation for the loss queue within the network is similar to that of an isolated facility when vehicles in the facility are permitted into downstream facilities or outside the intersection (i.e.,$\sum_{j \in \mathcal{N} \cup \{0\}} pr_{ij}(t) = 1$).

The output-flow rate of the feedback queue $\theta_i^F(t)$ within the network has a new form

$$\theta_i^F(t) = u_i'(t) \left(1 - PE_i^F\left(x_i^F(t)\right)\right) \sum_{j \in \mathcal{N} \cup \{0\}} pr_{ij}(t), \tag{30}$$

$$u_i'(t) = \sum_{j \in \mathcal{N} \cup \{0\}} \left[\left(u_i(x_i^F(t))pr_{ij}(t)\right)^{-1} + PB_j^L\left(x_j^L(t)\right)\sigma_j'(t)^{-1}\right]^{-1} \tag{31}$$

where the service rate $u_i(x_i^F(t))$ and the probability of zero vehicles $PE_i^F\left(x_i^F(t)\right)$ are still formulated by Equations (6) and (17)respectively. $u_i'(t)$ is the new service rate of facility $i$ considering service times of artificial nodes at downstream facilities. It models congestion propagation (feedback queues) from downstream facilities. Fig 7a)-c) shows the process of updating service rates in three-node feedback queuing network. Fig 7a) is the network reconfiguration stage where the artificial node $h_i$ with service rate $\sigma_i$ is added for each facility to register blocked vehicles. The artificial node is an $M_i(t)/G_i/\infty$ fluid queue. Fig 7b) is the feedback elimination stage. The repeated visits (feedback) to the artificial node create strong dependencies in arrival processes. To avoid such dependencies, a reconfiguration of the artificial node needs to be performed. The feedback arc is removed from the old artificial node, and the new artificial node $h_i'$ is an $M_i(t)/G_i/1$ fluid queue with new service rate $\sigma_i'(t)$. Fig 7c) is the artificial node removal stage. To solve the output-flow rate of the feedback queue, the facility and the artificial nodes at downstream facilities need to be combined by Equations (30) and (31). The equation (31) shows that the actual mean service time of the facility $u_i'(t)^{-1}$ is the mean service time before congestion propagation plus the service times of new artificial nodes at congested downstream facilities. This idea is inspired by the generalized expansion methods (GEM) [81].

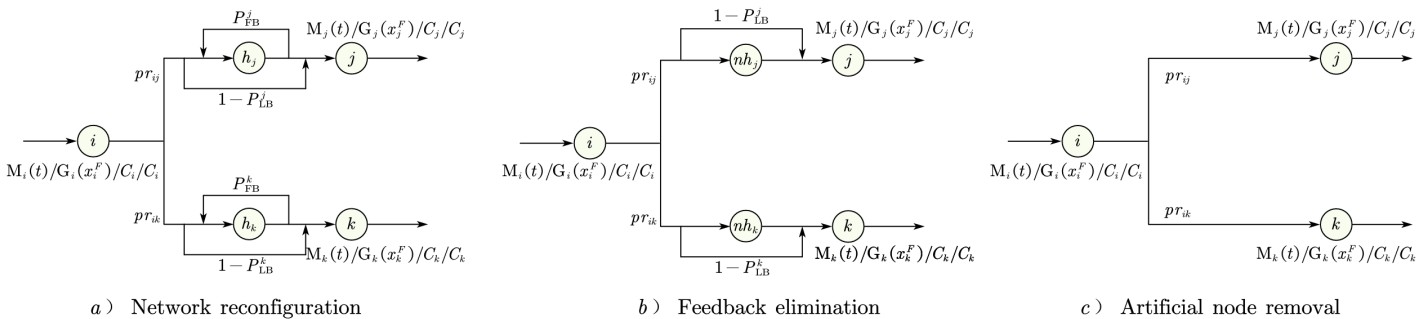

a）Network reconfiguration　　　　　b）Feedback elimination　　　　　c）Artificial node removal

**Fig 7. Process of updating service rates in a three-node feedback queuing network.**

Finally, according to the flow conservation, we have

$$\lambda_i^L(t) = \lambda_{0i}(t) + \sum_{j \in \mathcal{N}} \theta_j^L(t) pr_{ji}(t),$$

(32)

which indicates that the effective arrival rate for each loss system $\lambda_i^L(t), i \in \mathcal{N}$ consists of the external arrival rate $\lambda_{0i}(t), i \in \mathcal{N}$ and the departure rates of upstream loss systems.

Thus, Equations (27)-(32) formulate the time-varying and state-dependent feedback queuing network model for the signalized intersection.

Following the aforementioned methodology, an intersection queuing network model can be constructed, abstracting signalized intersections into a network of interconnected queuing nodes. Consistent with this framework, multi-intersection clusters can likewise be represented as queuing networks composed of multiple nodes. Consequently, the queuing network model for intersection clusters established in this study can be decomposed into multiple isolated intersections, enabling parallel optimization of signal timing plans for clustered intersections.

Furthermore, our prior research [56] has demonstrated that the computational complexity of the recursive algorithm for solving the intersection-cluster queuing network model is $O\left(N^2 \Delta T\right) + O(N \Delta T)$, where $N$ denotes the number of intersection service facilities and $\Delta T$ the number of discrete time intervals. When addressing short-term horizons (e.g., 10-minute periods), the transition probabilities can be assumed constant, further reducing complexity to $O(T) + O(NT)$. This low-complexity characteristic endows the algorithm with significant potential for large-scale network applications.

## Performance measures

By solving Equations (27)-(32), we can obtain the average number of vehicles $x_i^F(t)$ (system state) at any time $t$ for any facility $i$. We can then calculate the dynamic performance measures of interest such as the output-flow rate (throughput) $\theta_i^F(t)$ at any time $t$, the arrival rate $\lambda_i(t)$ at any time $t$, the vehicle velocity $v_i\left(x_i^F(t)\right)$ at any time $t$, and the vehicle blocking probability $PB_i^L\left(x_i^L(t)\right)$ at any time $t$ by Equations (30), (32), (3) and (10). In Section 6, we develop a recursive algorithm to solve the queuing network.

## Optimization Frameworks

In this section, we present multiple optimization frameworks which solve the optimal control of traffic signals for the signalized intersection. These optimization frameworks can be ultimately formulated by substituting the performance measures of the proposed feedback queuing network model. To compare with the famous traffic signal timing software Synchro in numerical experiments, the basic optimization framework is similar to that of Synchro in subsection 5.1, which minimizes the average delay of the intersection by constraining the cycle time and the green times for different phases. We have an insight into how congestion propagation in dynamic stochastic environments affects traffic signal control. In subsection 5.2, we present other optimization frameworks in which our objective is extended to minimize the total costs of delay time and fuel consumption or maximize the system throughput. Besides, we also discuss how to extend our model to solve the traffic signal control of multiple signalized intersections in incomplete connected automated vehicle environments.

## Basic model

Our basic optimization framework can be formulated:

**Objective function: Minimizing the average delay.**

$$\min Z = \frac{\sum_{j \in \{E,S,W,N\}} D_j(t_0, t_0 + T)}{\sum_{j \in \{E,S,W,N\}} d_j(t_0, t_0 + T)},$$

(33)

where $t_0$ is the initial time of the rolling optimization, and $T$ is the corresponding time step. Generally, $t_0$ is the start time of a signal cycle. Here, $T$ is not required to be an integral multiple of the signal cycle length. $D_j(t_0, t_0 + T)$ is the total delay time of all vehicles during the rolling time domain $[t_0, t_0 + T]$ in direction $j \in \{E, S, W, N\}$. $d_j(t_0, t_0 + T)$ is the corresponding number of arrival vehicles. $D_j(t_0, t_0 + T)$ and $d_j(t_0, t_0 + T)$ can be obtained by the performance measures of the proposed queueing network model, i.e.,

$$D_j(t_0, t_0 + T) = \sum_{i \in \mathcal{N}_j \cap \mathcal{N}_1} \int_{t_0}^{t_0+T} y_i(t) dt + \sum_{i \in \mathcal{N}_j - \mathcal{N}_3} \int_{t_0}^{t_0+T} x_i^F(t) dt, \forall j \in \{E, S, W, N\}, \tag{34}$$

Here, the first item on the right of Equation (34) is the total delay time of vehicles overflowing outside the imported road sections. The second item is the total delay time of vehicles inside the imported road sections and entrance lanes.

$$d_j(t_0, t_0 + T) = \sum_{i \in \mathcal{N}_j \cap \mathcal{N}_1} y_i(t_0) + \sum_{i \in \mathcal{N}_j - \mathcal{N}_3} x_i^F(t_0) + \sum_{i \in \mathcal{N}_j \cap \mathcal{N}_1} \int_{t_0}^{t_0+T} \lambda_{0i}(t) dt, \forall j \in \{E, S, W, N\}, \tag{35}$$

Here, the first two items on the right of Equation (35) is the total number of vehicles which already existed at the initial time. The third item is the total number of new arrival vehicles at imported road sections during the rolling time domain.

To fairly compare with Synchro software in the same optimization framework, this basic objective function does not consider the delay of exported road sections. However, Formula (33) can capture the congestion propagation from exported road segments on the traffic signal control. This is because the average number of vehicles $x_i^F(t)$ and $y_i(t)$ are obtained by the proposed feedback queuing network model. In the following subsection, we also present multiple optimization objective functions of interest, which can be all ultimately formulated based on the proposed feedback queuing network model.

**Green time constraints.**
$$g_k^{\min} \leq g_k \leq g_k^{\max}, \forall k \in \mathcal{K}, \tag{36}$$

where $g_k$ is the green time at signal phase $k$ in the rolling optimization time domain $[t_0, t_0 + T]$. Input parameters $g_k^{\min}$ and $g_k^{\max}$ are the minimum and maximum green times, respectively. $g_k, \forall k \in \mathcal{K}$ are decision variables that will be updated by the rolling optimization time interval.

**Cycle time constraints.**
$$\sum_{k \in \mathcal{K}} (g_k + y_k + r_k) = CL, \tag{37}$$

$$CL^{\min} \leq CL \leq CL^{\max}, \tag{38}$$

where input parameters $y_k$ and $r_k$ are the yellow- and all-red times at signal phase $k \in \mathcal{K}$. The signal cycle time $CL$ is also a decision variable. It will be updated with the rolling optimization time interval $[t_0, t_0 + T]$. The equation (37) is the general expression for the cycle time. Input parameters $CL^{\min}$ and $CL^{\max}$ are the minimum and maximum cycle times, respectively.

**Integer constraints.**
$$g_k, CL \in \mathbb{Z}, \forall k \in \mathcal{K}. \tag{39}$$

Generally, the green time and cycle time are integers in the real world. That is, our model is pure integer programming.

## Model extensions

It is worth noting that although the basic model aims to minimize the average delay of the intersection, it can be easily extended to optimize other objective functions. Recently, green traffic, energy conservation and environmental protection have received much more attention. Fuel consumption has been taken into account during the traffic signal control by many researchers [82,83]. Our model can also incorporate the fuel consumption or emission, which is also affected by the congestion propagation, into the following objective function.

## Minimizing the average total costs

$$\min Z = Z_1 + Z_2. \tag{40}$$

where $Z_1$ and $Z_2$ means the average delay time costs and the average fuel consumption or emission costs, respectively. The average delay time costs $Z_1$ for a vehicle in the rolling optimization time interval $[t_0, t_0 + T]$ have the following forms.

$$Z_1 = w_1 \frac{\sum_{j \in \{E,S,W,N\}} D_j(t_0, t_0 + T)}{\sum_{j \in \{E,S,W,N\}} d_j(t_0, t_0 + T)}, \tag{41}$$

where $w_1$ is the unit costs per unit delay time per vehicle. The unit costs of delay can be determined by the average salary and working hours of urban residents. And the $D_j(t_0, t_0 + T)$ and $d_j(t_0, t_0 + T)$ can be obtained by equation (34)and equation (35).

The average Fuel Consumption (FC) or Emission $(CO, HC, NO_x)$ costs $Z_2$ for a vehicle in the rolling optimization time interval $[t_0, t_0 + T]$ have the following forms.

$$Z_2 = w_2 \frac{\sum_{j \in \{E,S,W,N\}} F_j^z(t_0, t_0 + T)}{\sum_{j \in \{E,S,W,N\}} d_j(t_0, t_0 + T)}, \tag{42}$$

Where, $w_2$ is the unit costs of emission or fuel consumption which can be determined by the unit governance cost of emission or gasoline price of a city. $F_j^z(t_0, t_0 + T)$ is the total emission $(CO, HC, NO_x)$ or fuel consumption (FC) of all vehicles during the rolling time domain $[t_0, t_0 + T]$ in direction $j \in \{E, S, W, N\}$. $d_j(t_0, t_0 + T)$ is the corresponding number of arrival vehicles, can be obtained by equation (35). $F_j^z(t_0, t_0 + T)$ can be expressed by:

$$F_j^z(t_0, t_0 + T) = \sum_{i \in \mathcal{N}_j \cap \mathcal{N}_1} \int_{t_0}^{t_0+T} f_i^z(t) y_i(t) \mathrm{d}t + \sum_{i \in \mathcal{N}_j - \mathcal{N}_3} \int_{t_0}^{t_0+T} f_i^z(t) x_i^F(t) \mathrm{d}t, \forall j \in \{E, S, W, N\}, \tag{43}$$

Where, $f_i^z(t)$ is the macroscopic emission and fuel consumption for the vehicles moving within a section. Following the macroscopic emission and fuel consumption model proposed by [84,85], $f_i^z(t)$ is obtained as:

$$f_i^z(t) = T_s x_i^F(t) \exp(v_i(t) P_z a_i(t)), \tag{44}$$

where $f_i^z(t), z \in \{CO, HC, NO_x, FC\}$ denotes the average emission $(CO, HC, NO_x)$ or fuel consumption (FC) of node $i$ at time $t$; $T_s$ is the simulation step in this paper $T_s = 1s$. $x_i^F(t)$ represent the system state of node $i$ at time $t$, which can be obtained by Equation (27); $v_i(t)$ is the average velocity of node $i$ at time $t$, which can be obtain by Equation (3); and $a_i(t)$ means the average acceleration of node $i$ at time $t$, which can be obtained by

$$a_i(t) = \frac{\mathrm{d}v_i\left(x_i^F(t)\right)}{\mathrm{d}t},\tag{45}$$

$P_z$ is the parameter for the emission variable $z \in \{\mathrm{CO}, \mathrm{HC}, \mathrm{NO}_x\}$ or fuel consumption variable ($z = \mathrm{FC}$) which is given by:

$$P_{\mathrm{CO}} = 0.01 \begin{bmatrix} -1292.81 & 48.8324 & 32.8837 & -4.7675 \\ 23.2920 & 4.1656 & -3.2843 & 0 \\ -0.8503 & 0.3291 & 0.5700 & -0.0532 \\ 0.0163 & -0.0082 & -0.0118 & 0 \end{bmatrix},\tag{46}$$

$$P_{\mathrm{HC}} = 0.01 \begin{bmatrix} -1454.4 & 0 & 25.1563 & -0.3284 \\ 8.1857 & 10.9200 & -1.9423 & -1.2745 \\ -0.2260 & -0.3531 & 0.4356 & 0.1258 \\ 0.0069 & 0.0072 & -0.0080 & -0.0021 \end{bmatrix},\tag{47}$$

$$P_{\mathrm{NO}_x} = 0.01 \begin{bmatrix} -1488.32 & 83.4524 & 9.5433 & -3.3549 \\ 15.2306 & 16.6647 & 10.1565 & -3.7076 \\ -0.1830 & -0.4591 & -0.6836 & 0.0737 \\ 0.0020 & 0.0038 & 0.0091 & -0.0016 \end{bmatrix},\tag{48}$$

And

$$P_{\mathrm{FC}} = 0.01 \begin{bmatrix} -753.7 & 44.3809 & 17.1641 & -4.2024 \\ 9.7326 & 5.1753 & 0.2942 & -0.7068 \\ -0.3014 & -0.0742 & 0.0109 & 0.0116 \\ 0.0053 & 0.0006 & -0.0010 & -0.0006 \end{bmatrix}.\tag{49}$$

One can see [85] for the detailed calculation process of the emission or fuel consumption.

Some researchers focus on the throughput optimal traffic signal control [86,87]. We can also maximize the system throughput of the intersection by considering congestion propagation in dynamic stochastic environments. Besides, our model can be extended to solve the traffic signal control of multiple signalized intersections when the proposed feedback queuing network model in Section 4 uses the network topology of multiple intersections. Our model is also promising to handle the traffic signal control in incomplete connected automated vehicle environments if the feedback queuing network model considers two types of vehicles which is our ongoing work.

## Algorithm

In this section, we develop a recursive algorithm to solve the feedback queuing network model. The algorithm for solving the queuing network is nested within the optimization frameworks, optimizing the green times and cycle time in the rolling time interval. The optimization models presented in Section 5 are non-convex programming, and the mesh adaptive direct search algorithm (MADS) is used here. Meanwhile, a rolling optimization strategy embedding MADS is presented to realize the real-time traffic signal control.

## Queuing algorithm

To reduce the queuing network's computation complexity, we develop the following recursive algorithm where time is discretized into seconds. Different equations are approximately solved by the explicit Euler method (a NI method). Note that one can also use other more advanced NI methods such as implicit Euler or Runge-Kutta to yield better results.

**Algorithm 1.** A recursive algorithm for the feedback queuing network of intersections
**Input:** All system parameters required for the feedback queuing network model, e.g., system state of each facility at the initial time $(x_i^F(t_0) := x_{i0}; x_i^L(t_0) := x_{i0})$, transition probability of potential traffic demand $pc_{ij}(t)$, external arrival rate $\lambda_{0i}(t)$, dimension of each facility ($l_i$, $w_i$), and traffic signal timing plan ($g_k$, $y_k$, $r_k$, $CL$) where $\forall i, j \in \mathcal{N}, k \in \mathcal{K}, t \in [t_0, t_0 + T]$.
**Output:** All performance measures of interest for the feedback queuing network model at any time, e.g., system state $x_i^F(t)$, throughput $\theta_i^F(t)$, vehicle velocity $v_i(x_i^F(t))$, and vehicle acceleration $A_i(t)$ where $\forall i \in \mathcal{N}, t \in [t_0, t_0 + T]$
**Recursive Computation for Queueing Network**
**For** $i \in \mathcal{N}$
 Compute $C_i, r_i(C_i), v_i(C_i), \mu_i(C_i), \sigma_i$ in sequence by Equations,,, and
**End For**
**For** $t = t_0, t_0 + 1, \ldots, t_0 + T - 1$
 **For** $i \in \mathcal{N}$
 **For** $j \in \mathcal{N}$
 Compute $a_{ij}(t)$, $pr_{ij}(t)$ by Equation(4)
 **End For**
 Compute $PB_i^L$, $PE_i^F$, $PB_i^F$ in sequence by Equations(10), (17) and(22)
 **End For**
 **For** $i \in \mathcal{N}$
 Compute $u_i'(t), \theta_i^F(t)$ in sequence by Equations (31) and(30)
 **End For**
 **For** $i \in \mathcal{N}$
 Compute $\lambda_i(t), PE_i^L, \theta_i^L(t)$ by Equations(5),(9),(29)
 Compute $x_i^L(t+1)$ after discretizing Equation (28) by $\frac{dx_i^L(t)}{dt} \cong x_i^L(t+1) - x_i^L(t)$
 Compute $x_i^F(t+1)$ after discretizing Equation(27) by $\frac{dx_i^F(t)}{dt} \cong x_i^F(t+1) - x_i^F(t)$
 **End For**
**End For**

Because of the computation for time-varying transition probability $pr_{ij}(t)$ in Equation (4), the proposed recursive algorithm in Algorithm 1 has the complexity of $O(N^2 T) + O(NT)$ where $N$ is the number of facilities. If time step of rolling optimization $T$ is small (e.g., 5 minutes), one can consider transition probability of potential traffic demand $pc_{ij}(t)$ as constant in the rolling time interval $[t_0, t_0 + T]$. So that, the complexity can be reduced to $O(N_2 N_3 T) + O(NT)$ where $N_2$ and $N_3$ is respectively the numbers of entrance lanes and exported road sections.

## Optimization algorithm

The algorithm for solving the queuing network behavior in Section 6.1 is nested into optimization frameworks in Section 5 to complete our optimization models. Our optimization models belong to a non-convex integer programming which is generally an NP-Hard problem. By plotting figures, we find that our objective functions are non-convex and non-concave. The performance measures for the feedback queuing network model (include system state $x_i^F(t)$, throughput $\theta_i^F(t)$, vehicle velocity $v_i(x_i^F(t))$, and vehicle acceleration $A_i(t)$) are nonlinear functions with no closed-form (solved by a recursive algorithm). By omitting the integer constraints, the continuous relaxation of the non-convex integer programming is a global optimization problem that is very difficult to solve. To solve the optimization problem, we use the MADS algorithm proposed by [88]. It can effectively solve global nonlinear non-convex mixed-integer programming for problems up to a few hundred variables [89]. To realize the real-time traffic signal control, the MADS algorithm is nested into the rolling optimization strategy as follows.

**Algorithm 2.** Rolling optimization for the traffic signal control of intersections
**Step 1:** Input initial time $t_0$ and rolling time step $T$
**Step 2:** Predict external arrival rate $\lambda_{0i}(t)$ and transition probability of potential traffic demand $pc_{ij}(t)$ during the rolling time interval $[t_0, t_0 + T]$, and obtain the system state of each facility $x_{i0}$ at the initial time based on data collected in real-time

```
Step 3: Solve the optimization model and obtain the signal control scheme using MADS
```
**Step 4:** If $T - \lfloor\frac{T}{CL}\rfloor CL \geq \frac{CL}{2}$, execute the signal control scheme in the time interval $[t_0, t_0 + \lceil\frac{T}{CL}\rceil CL]$ and update initial time $t_0 := t_0 + \lceil\frac{T}{CL}\rceil CL$. Otherwise, execute the signal control scheme in the time interval $[t_0, t_0 + \lfloor\frac{T}{CL}\rfloor CL]$ and update initial time $t_0 := t_0 + \lfloor\frac{T}{CL}\rfloor CL$. Then go back to Step 1

In Step 2 of Algorithm 2, one can obtain system parameters required for the rolling optimization in real-time by GPS data and prediction techniques (e.g., [90–92]),which is not our focus in this paper. These system parameters are given in advance in numerical experiments. In Step 4, the signal control scheme in a cycle is fully executed by updating the end time.

## Numerical Experiments

We firstly verify our queueing network model in section 6.1. After that, we take an actual intersection in Kunshan as the case study in section 6.2. According to the proposed models and algorithms, all the tests are solved by the general solver MATLAB R2019a. The tests are run on a Studio PC with 1.6 Gigahertz of Intel(R) core i5-8265U CPU and 8 Gigabyte of RAM in a Windows environment.

## Queuing model verification

We first explain how we built the base scenarios using real data collect in Kunshan city. The signalized intersection of Changjiang and Qianjin middle roads is considered, as shown in Fig 8. The speed limit and jam density for each road are 60 km/h and 160 veh/(km-lane), respectively. From Table 3, we take the three representative points $v_0$, $v_a$ and $v_b$ as 60, 20, and 5 km/h, respectively, to calibrate the velocity model. Thus, each lane's maximum traffic capacity is 1650 veh/h from Equation(3), which is in line with the on-ground observation. The observed data for the 15-minutes traffic demand on imported road sections are shown in Table 4. We use the cubic polynomial function to fit the external traffic demand $\lambda_{0i}(t), i \in \mathcal{N}_1, t \in [17:00, 18:00]$. The initial system state is set as 0 vehicles. The other survey data is given in Table 5.

Then, we verify the proposed feedback queueing network model through discrete event simulation experiments under three demand scenarios with different traffic intensities. We refer to the demand in Table 4 as the basic scenarios. Table 6 gives the three demand scenarios considered in this section. The demand scenarios 1~3 represent the unsaturation traffic intensity, saturation traffic intensity, and oversaturation traffic intensity, respectively. The first column values represent the number of times of the demand in each scenario concerning the base scenarios' demand. For example, the generated traffic demand in the second scenario is traffic demand in the basic scenarios multiplied by 2. The generated traffic demand in the first scenario is just traffic demand in the basic scenarios. The values from the second to fifth columns represent the maximum traffic intensity in each direction. The values from the sixth to ninth columns represent the average traffic intensity.

We establish a discrete event simulation model for the signalized intersection using the Simulink toolbox in MATLAB R2019a. The above three demand scenarios are tested. Each simulation group is repeated 200 times with random seeds. Figs 9–11 shows the dynamic system state $x(t)$ under the three demand scenarios, which are calculated by the queuing model and the simulation model. The "simulation" in Figs 9–11 is the mean result of the 200 simulations. The "confidence interval" in Fig 9–11 is the 95% confidence interval of the 200 simulations.

Figs 9–11 shows that the difference between the simulation model results and those of the proposed model is tiny regardless of the variance in traffic intensity. Compared with the mean result of the 200 simulations, the average absolute error is 0.5152 vehicles, and the average relative error is 6.43%. Figs 9–11 shows that the proposed model can capture the dynamic performance measures of the intersection accurately. Therefore, the proposed $\mathbf{M}_t/G(x)/C/C$ feedback queueing network model accurately describes the dynamic performance of the intersection. It should be noted that each simulation takes nearly 0.5 hours. The proposed model only takes around 0.2 seconds, although the approach's time step is set to 1 second, and the period [17:00, 18:00) is divided into 3600 segments. This indicates that the proposed recursive

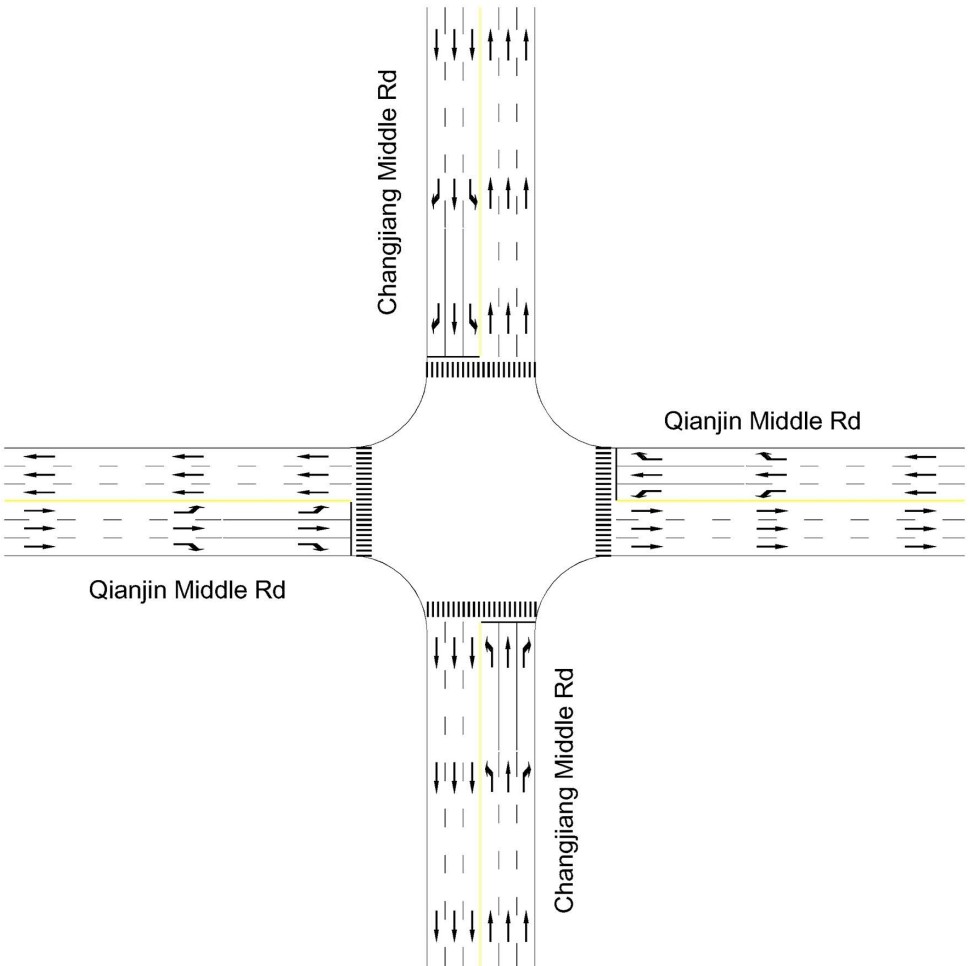

**Fig 8. Signalized intersection in Kunshan.**

**Table 4. Observed data for 15-minutes traffic demand on imported road sections.**

|  | East | South | West | North |
|---|---|---|---|---|
| 17:00-17:15 | 129 | 121 | 101 | 54 |
| 17:15-17:30 | 209 | 116 | 114 | 32 |
| 17:30-17:45 | 155 | 122 | 87 | 70 |
| 17:45-18:00 | 177 | 72 | 161 | 71 |

algorithm in Algorithm 1 has high computational efficiency for the intersection queue network system. It is promising to solve large-scale problems such as 200 intersections which is our ongoing work.

## Basic optimization model analysis

In this subsection, we first compare the signal timing results between the proposed basic optimization model and the famous Synchro software in different demand and initial state scenarios. Then, we analyze the impact of rolling optimization strategy on the proposed basic optimization model's signal timing results. Both of them have a similar optimization framework

**Table 5. Survey data of the intersection.**

| | | Import road section | Entrance lane | | | Exported road section |
|---|---|---|---|---|---|---|
| | | | Left-turn | Straight | Right-turn | |
| East | Length (m) | 285 | 50 | 50 | 50 | 335 |
| | Number of lanes | 4 | 1 | 2 | 1 | 4 |
| | Path probability (%) | - | 16 | 70 | 14 | - |
| South | Length (m) | 483 | 70 | 70 | 70 | 553 |
| | Number of lanes | 3 | 1 | 1 | 1 | 3 |
| | Path probability (%) | - | 47 | 38 | 16 | - |
| West | Length (m) | 253 | 50 | 50 | 50 | 303 |
| | Number of lanes | 4 | 1 | 2 | 1 | 4 |
| | Path probability (%) | - | 26 | 65 | 9 | - |
| North | Length (m) | 280 | 70 | 70 | 70 | 350 |
| | Number of lanes | 3 | 1 | 1 | 1 | 3 |
| | Path probability (%) | - | 42 | 35 | 23 | - |

**Table 6. Demand scenarios from 17:00 to 18:00 in simulation verification.**

| Demand scenarios | The maximum traffic intensity | | | | The average traffic intensity | | | |
|---|---|---|---|---|---|---|---|---|
| | East | South | West | North | East | South | West | North |
| 1 | 0.5198 | 0.5926 | 0.2487 | 0.5479 | 0.4184 | 0.4574 | 0.1851 | 0.3109 |
| 2 | 1.0396 | 1.1852 | 0.4973 | 1.0957 | 0.8368 | 0.9148 | 0.3702 | 0.6218 |
| 3 | 1.5594 | 1.7778 | 0.7461 | 1.6437 | 1.2552 | 1.3722 | 0.5553 | 0.9327 |

(objective and constraints). Our basic optimization model calculates the average delay by the proposed $M_t/G(x)/C/C$ feedback queueing network model which considers the congestion propagation in dynamic stochastic environments. In comparison, the Synchro software calculates the average delay by the Webster model [1], which does not consider congestion propagation. Therefore, we can analyze the impact of the congestion propagation on traffic signal control.

## Comparison in different demand and initial state scenarios

Fig 12 a) – c) shows results for the proposed model and Synchro software in different combinations of demand and initial states. Here, both of the two approaches do not use the rolling optimization strategy. The initial states are set as 0.1, 0.3, 0.5, 0.7 and 0.9 times of the facility capacity which correspond to different traffic densities. The demand scenarios are set as 0.5, 1.0, 1.5, 2.0, 2.5 and 3.0 times of the basic demand scenario. Fig 12a) displays the optimal average delay of the proposed basic approach. Fig 12b) presents the percentage of delay reduced by the proposed basic approach compared with the Synchro software. Fig 12c) gives the relative change of green time for the four signal phases. In Fig 12c), 30 traffic scenarios for the combinations of initial states and demand are sorted in lexicographic order, i.e., (0.1, 0.5), (0.1, 1.0) …., (0.9, 3.0).

Fig 12 a) shows that the optimal vehicle's average delay time exponentially increases with the initial state and demand scenarios. For example, we set the demand scenario as 0.5, and then the optimal average delay time of vehicle is 40.26s in initial state scenario 0.1. While it is up to 368.83s in initial state scenario 0.9. Set the initial state scenario as 0.3, and then the optimal average delay time of vehicle is 75.35s in demand scenario 0.5. And it is 1913.47s under demand scenario 3.0. This is because the traffic intensity and CPDSE in the intersection increases with the initial state and demand scenarios. The vehicle velocity exponentially decreases with the growing traffic intensity and CPDSE in our proposed queueing model (see Fig 3 (a)).

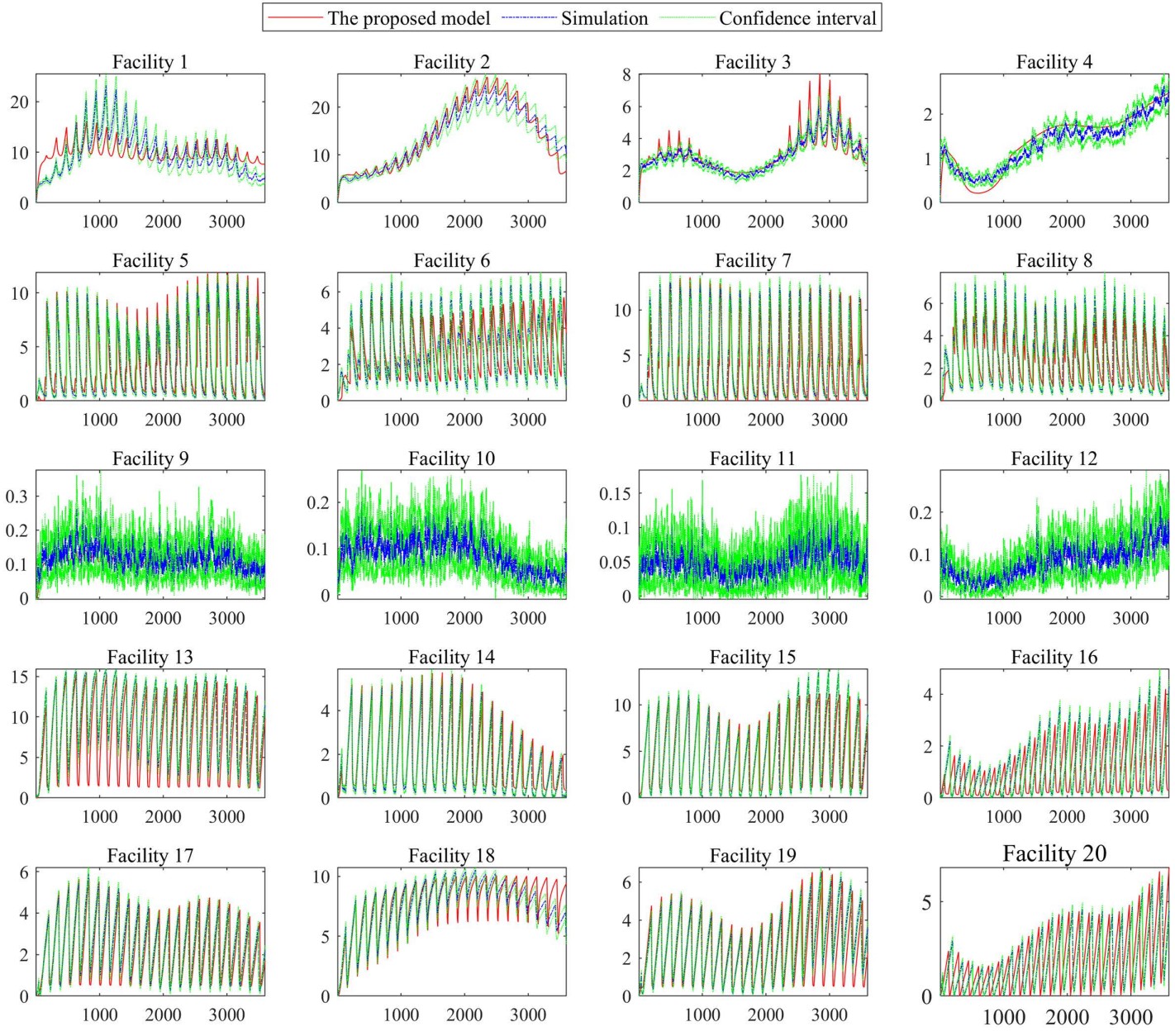

**Fig 9. Dynamic system state in demand scenarios 1.**

Fig 12b) shows the percentage of delay reduced by our method compared with the Synchro software. There are some interesting findings:

(1) In the lowest traffic scenario (0.1, 0.5), the percentages of reduced delay is vary small (0.1%). That is, the optimal vehicle's average delay time of our model is approximate to that of Synchro in the lowest traffic scenarios. This is because that the CPDSE is weak in the lowest traffic scenario. Our model gradually degrades to the Synchro model.

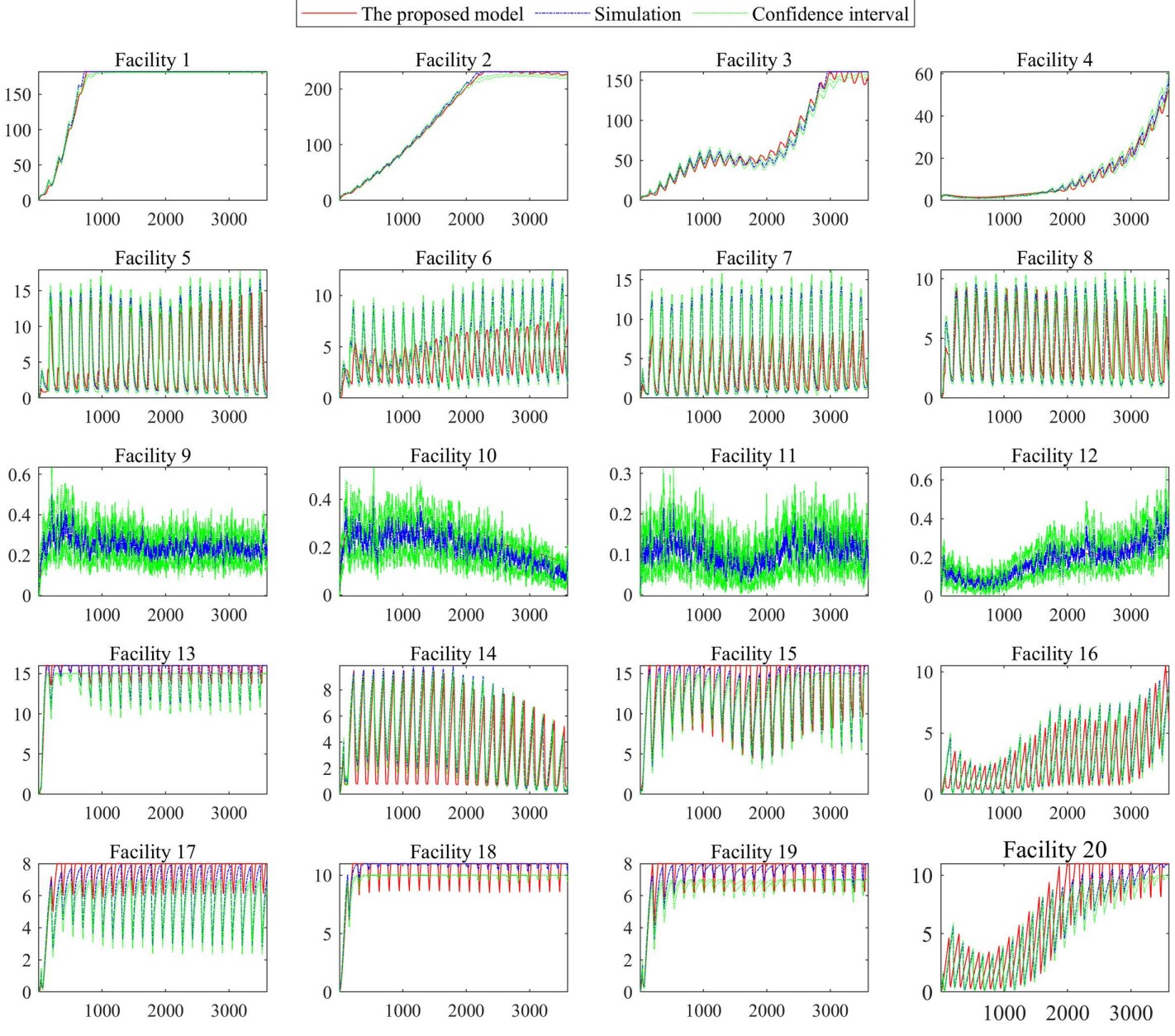

**Fig 10. Dynamic system state in demand scenarios 2.**

(2) The percentage of reduced delay reaches the maximum (64.98%) in moderate traffic scenario (0.3, 1.0). In this scenario, the initial state is moderately crowded and the traffic demand is small. As our queueing model considers the CPDSE and exponentially descending velocity, the moderate traffic congestion at the initial moment is difficult to dissipate immediately, but it can dissipate in a right time period due to the low traffic demand. Our optimization model can find the best time period to dissipate the moderate traffic congestion as soon as possible. However, Synchro does not consider these elements and its optimal signal timing has nothing to do with the initial state. That is, Synchro ignores the impact of initial states on queue dissipation. Therefore, compared with the Synchro software, our model can deeply reduce the vehicle's average delay time in this scenario.

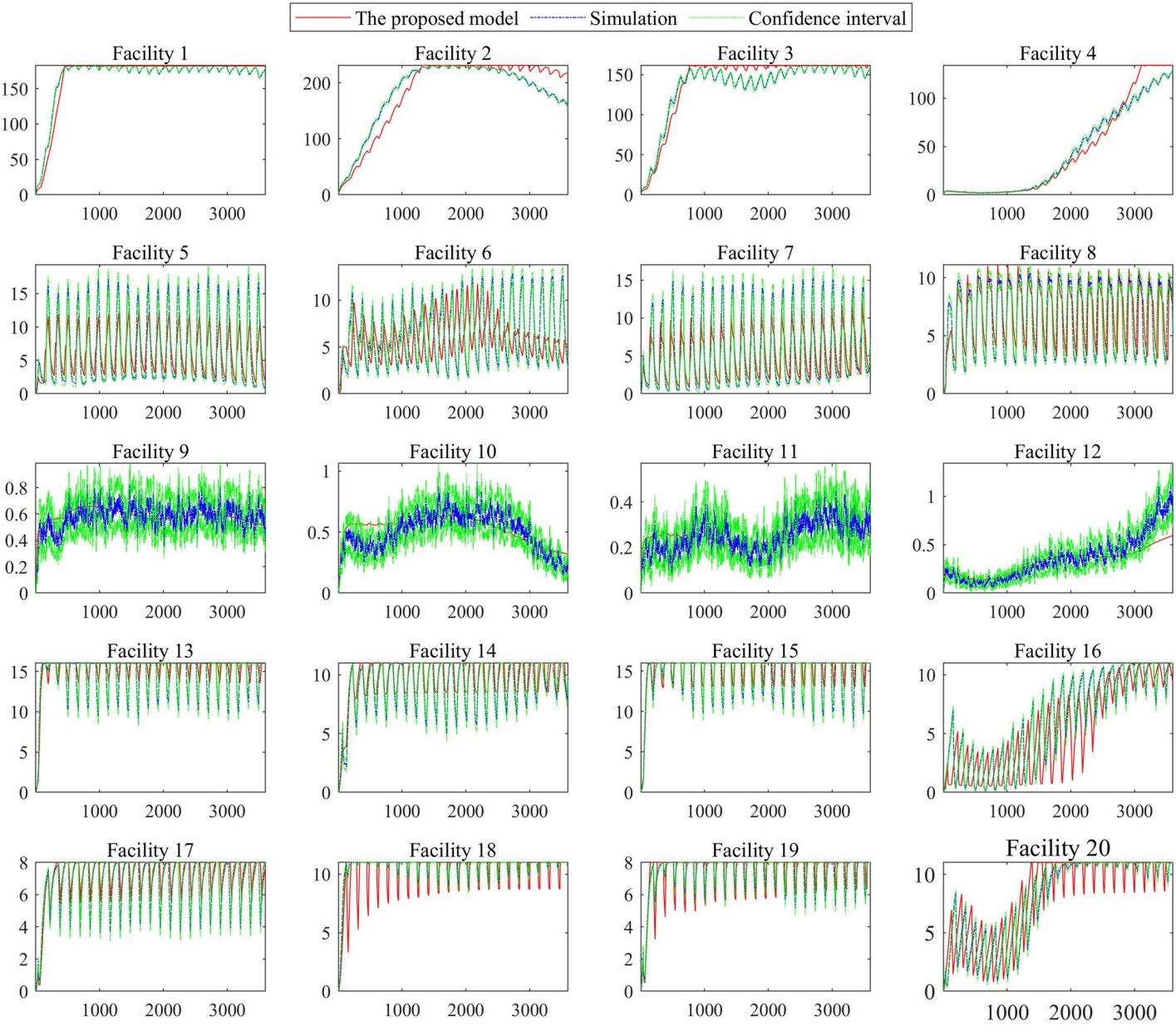

**Fig 11. Dynamic system state in demand scenarios 3.**

(3) In the lowest initial state scenario 0.1, the percentage of reduced delay climbs up and then declines with the demand. That is, there exists a moderate demand scenario (1.5) to maximize the gap (53.41%) between our model and Synchro. This is because our model gradually degrades to the Synchro model in the lower traffic scenario (0.1, 0.5) from the analysis above. In the moderate traffic scenario (0.1, 1.5), the intersection easily produce moderate traffic congestion and obvious CPDSE. Compared with the Synchro, our model can better optimize the vehicle's average delay time by reducing the CPDSE. In the higher (oversaturated) traffic scenario (0.1, 3.0), both our model and Synchro are weak to reduce the traffic congestion and CPDSE.

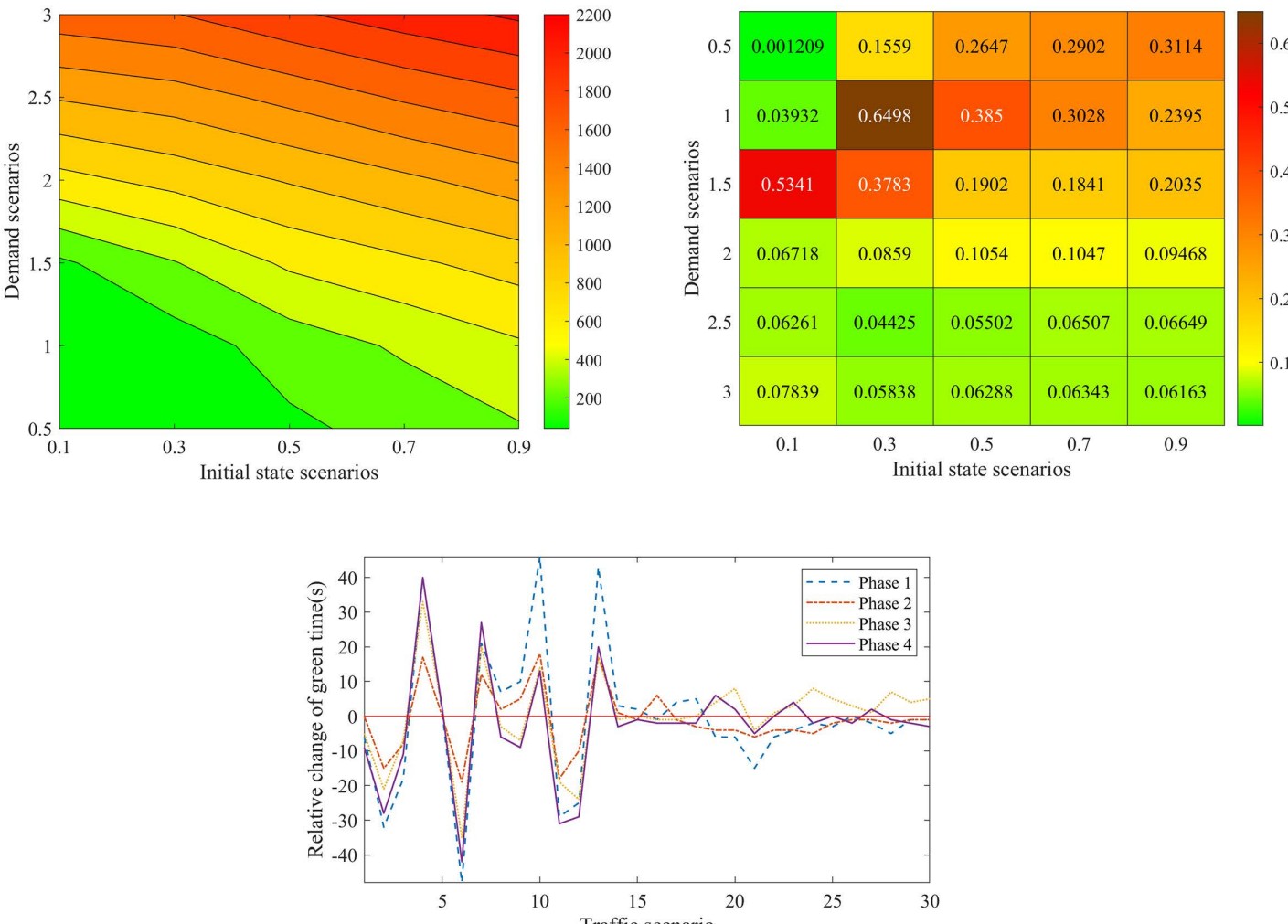

**Fig 12. Comparison in different demand and initial state scenarios. a Optimal average delay of the proposed basic approach. b Percentage of delay reduced by our method compared with the Synchro software. c.** Relative change of green time for the four signal phases method compared with the Synchro software.

(4) In the lowest demand scenario 0.5, the percentage of reduced delay shows a trend of increasing with initial state. This is because our model gradually degrades to the Synchro model in the lower traffic scenario (0.1, 0.5) from the analysis above. With the lowest demand and the moderate initial state, the intersection easily produce moderate traffic congestion and obvious CPDSE. As the initial state increases, the traffic congestion lasts longer. Compared with the Synchro, our model can better optimize the vehicle's average delay time by reducing the CPDSE.

(5) In higher demand scenarios (1.5~3.0) or higher initial state scenarios (0.7~0.9), the percentage of reduced delay decreases with the growing initial state or traffic demand. This is because that the traffic intensity becomes larger and larger. It is increasingly difficult to reduce the traffic congestion and the CPDSE.

Fig 12c) is the change of green time for the four signal phases compared with Synchro. Some interesting findings are revealed:

(1) The green time of each phase shows the same trend of first increasing and then decreasing with the growing initial states. This is because that in lower initial state scenarios (0.1~0.3), the traffic congestion at initial moment is slight. Our model increases the green time to dissipate the CPDSE quickly. In higher initial state scenarios (0.5~0.9), with the increase of traffic congestion at initial moment (i.e., the increase of initial states), it originally needs to take a longer green time to dissipate the CPDSE. However, unfortunately, the longer green time for one phase will result in the more CPDSE and waiting time at the other phases. Therefore, our model decreases the green time to balance CPDSE and waiting time at the other phases.

(2) The change of phase 1 is more than the other phases because the traffic intensity is more enormous than the other phase. The reduction of vehicle's average delay time of phase 1 affect the reduction of overall average delay time more obviously.

(3) The change of the green time is decreased as the demand scenarios increase. When the demand scenarios is high, (2.0–3.0), the traffic congestion at initial moment is serious. It takes a longer green time to dissipate the traffic congestion. However, one phase means the more waiting time at other phases, unfortunately. Therefore, it is difficult to reduce the delay time of vehicles by changing the green time.

## Comparison with existing methods

(1) Comparison with Osorio's method

The proposed method in this paper shares certain similarities with the approach by Osorio [60].Therefore, a comparison with their method is conducted to further demonstrate the effectiveness of our model. Osorio [60] employed an analytical $M(t)/M/1/C$ queuing network model to characterize traffic flow at intersections. However, their model abstracts the roadway link as a queuing network with a single server, fails to distinguish and characterize the traffic flow properties of different facilities, and does not account for the state-dependent nature of facility service capacity.

The optimization results obtained using the method proposed by Osorio [60] under different demand scenarios and initial state conditions are presented in Fig 13.

Compared to the method proposed by Osorio [60], our approach consistently achieves a better optimal average vehicle delay time, regardless of the initial system state or traffic demand scenario. This is because the method by Osorio [60] fails to account for the state-dependent service rate. It invariably provides an overly optimistic estimate of the intersection system's output rate, leading to excessive congestion in some facilities and consequently increasing vehicle delay times.

Under low traffic demand conditions, the optimal average vehicle delay times obtained by our proposed method and the method by Osorio [60] are very similar. This occurs because the probability of system congestion is low when traffic demand is small. Consequently, the assessments of the system output rate by both methods are relatively close, yielding similar results.

Compared to the method proposed by Osorio [60], the approach presented in this paper achieves further improvements in the optimal average delay time at intersections. The magnitude of these improvements is illustrated in Fig 14. The results demonstrate that compared to the method proposed by Osorio [60], the approach presented in this paper further reduces the optimal average delay time at intersections. The improvement is most significant under moderate traffic conditions, reaching a maximum reduction of 34.22%. This is because during moderate traffic conditions, the system experiences occasional congestion. Our model, which incorporates state-dependency, captures the system's dynamic performance more accurately, thereby enabling the optimization method to yield superior signal timing plans. Furthermore,

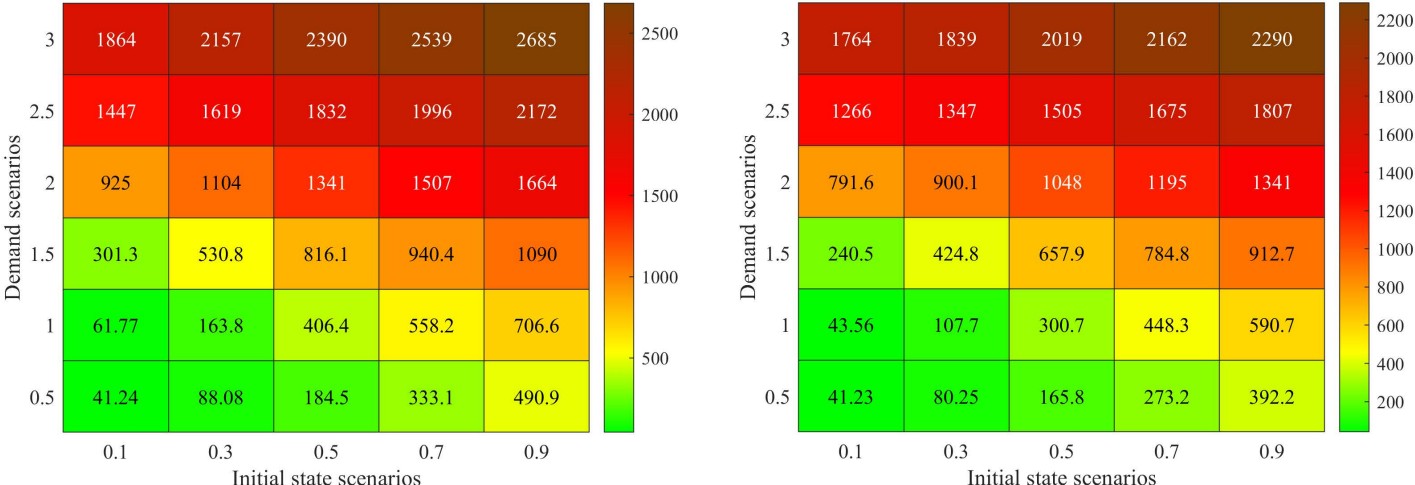

**Fig 13. Optimal average delay time of the existing method and the proposed model. (a)** The existing method. **(b)** The proposed method.

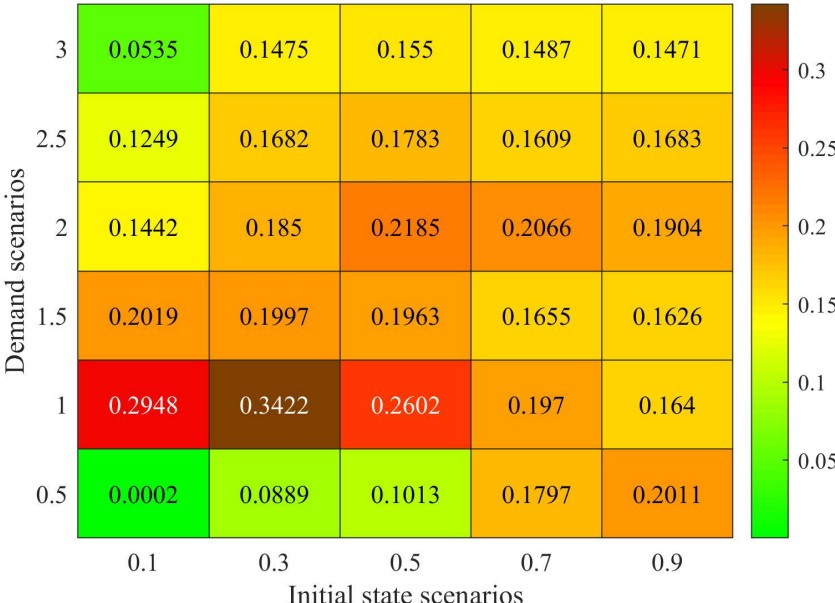

**Fig 14. The extent of improvement achieved by the proposed method over existing methods.**

the aforementioned results also indicate that, relative to the method by Osorio et al., our proposed method achieves further improvements in the optimal average intersection delay time across various traffic demands and initial states, albeit to varying degrees.

(2) Comparison with benchmark methods

In current research on traffic signal control, the SURTRAC [93] method and the MARLIN-ATSC [94] method represent landmark research outcomes. Therefore, this paper also conducts a comparative analysis with the aforementioned benchmark methods.

SURTRAC [93], developed by the research team at Carnegie Mellon University, is a decentralized real-time traffic signal control method with scalability at its core. Its fundamental principle lies in decomposing network-wide control into autonomous decision-making at individual intersections. Each intersection generates local signal schedules through rolling-horizon optimization and achieves predictive coordination with adjacent intersections via the sharing of lightweight "itinerary plans." The advantages of this method include exceptional computational efficiency, robust real-time adaptability, high robustness, and demonstrated effectiveness validated through field deployments (e.g., successfully reducing delays by over 40% in Pittsburgh). However, it is inherently a local optimization approach, which may not achieve global optimality, and its coordination efficacy relies on a stable communication network while being sensitive to the accuracy of sensor data. Despite these limitations, it remains a benchmark solution that successfully applies intelligent optimization theory to smart traffic management, combining practical utility with significant impact.

MARLIN-ATSC [94] is a pioneering multi-agent reinforcement learning framework for traffic control, developed by the research team at the University of Toronto. Its core concept enables each intersection agent to autonomously learn signal control policies through interaction with the environment, utilizing unique coordination mechanisms such as ICQ and DCQ to resolve competition and dependency issues among intersections. The method's prominent advantages lie in its powerful data-driven learning capability, which allows it to adapt to complex dynamic environments without predefined traffic models, and its significant potential to discover coordination strategies beyond traditional approaches. However, it is constrained by key challenges including prolonged training times due to high sample complexity, poor interpretability caused by the black-box nature of deep neural networks, and difficulties in ensuring the safety and stability of learned policies in real-world applications. These factors collectively limit its current potential for large-scale practical deployment and application.

Based on References [93] and [94], this study implemented both the SURTRAC and MARLIN-ATSC traffic signal control methods using Python to serve as benchmarks for comparison with the proposed method. The traffic demand and road network configurations remained consistent with those presented earlier in Tables 4 and Tables 5. Additionally, the initial system state was set at 0.3 times of the facility capacity. The results of the case study validation are summarized in Table 7.

Based on the comparative analysis and the data presented in Table 7, the proposed model demonstrates distinct and compelling advantages when evaluated against the two benchmarks, SURTRAC and MARLIN-ATSC.

In terms of control performance, both the proposed method and MARLIN-ATSC delivered excellent results, with very close average delay times (107.74 s and 103.62 s, respectively), and both outperformed SURTRAC (112.29 s). This indicates that all three methods are effective for intersection optimization, with MARLIN-ATSC's powerful data-driven learning capability enabling it to achieve the lowest delay under specific conditions.

However, the proposed model holds significant advantages in computational efficiency, interpretability, and practical applicability. As shown in Table 7, the computation time of the proposed method (48.1 s) is substantially lower than that of both SURTRAC (63.5 s) and MARLIN-ATSC (82.6 s), highlighting its superior potential for real-time applications. More importantly, the proposed model, grounded in explicit queuing theory, offers high interpretability, making its

Table 7. Performance Comparison between the Proposed Method, SURTRAC, and MARLIN-ATSC.

| Performance Indicators | Proposed Method | SURTRAC | MARLIN-ATSC |
|---|---|---|---|
| Average Delay Time (s) | 107.74 | 112.29 | 103.62 |
| Computation Time (s) | 48.1 | 63.5 | 82.6 |
| Interpretability | High | Moderate | Low |
| Scalability | High | High | Moderate |
| Data Dependency | Low | Moderate | Extremely high |

decision-making process transparent and trustworthy. In contrast, the "black-box" nature of MARLIN-ATSC's deep neural networks results in low interpretability, which hinders debugging and trust in practical deployments. Furthermore, the proposed model exhibits low data dependency, requiring only conventional traffic flow data. MARLIN-ATSC, conversely, relies on an extremely high volume and quality of training data, which severely constrains its applicability.

In summary, while MARLIN-ATSC can achieve marginally superior control performance when data is abundant, the proposed model offers a more balanced profile across computational efficiency, interpretability, and low data dependency. Compared to SURTRAC, the proposed model's explicit incorporation of congestion propagation and state-dependent characteristics, as opposed to their implicit handling within the prediction process, leads to slightly better performance. In conclusion, this research presents a promising solution for real-time traffic signal control that is not only effective and reliable but also highly practical for real-world deployment.

## Impact analysis of rolling optimization strategy

Figs 15–17 shows the proposed model's results with rolling optimization strategy in different combinations of demand and initial states.

Fig 15 is the proposed model's optimal average delay time under the rolling optimization strategy for different demand- and initial state scenarios. An exciting finding is that, there exists a optimal time step for rolling optimization to minimize the average delay time. (1) When the traffic demand (demand scenarios is 0.5~1.5) and initial state scenarios (0.1) are both slightly, the optimal average delay time in the given time period [17:00, 18:00) climbs up with the growing time step of the rolling optimization. The optimal step for rolling optimization is 300s. (2) When the traffic demand (demand scenarios is 2.0~3.0) is high and initial state scenarios (0.3~0.5) is moderate, the optimal average delay time in the given time period [17:00, 18:00) declines first and then climbs up with the growing time step of the rolling optimization. There is a

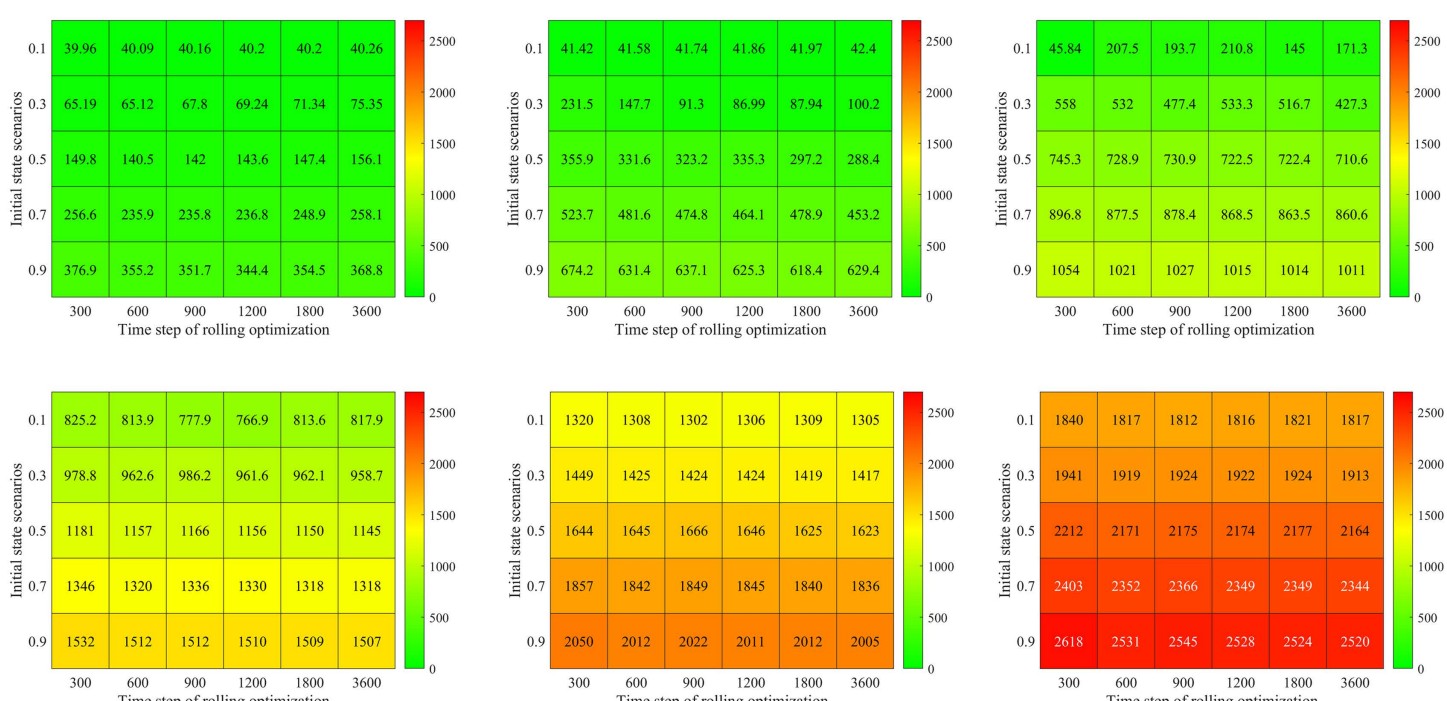

**Fig 15. Optimal average delay time of the proposed model with rolling optimization strategy. (a)** Demand scenarios 0.5. **(b)** Demand scenarios 1.0. **(c)** Demand scenarios 1.5. **(d)** Demand scenarios 2.0. **(e)** Demand scenarios 2.5. **(f)** Demand scenarios 3.0.

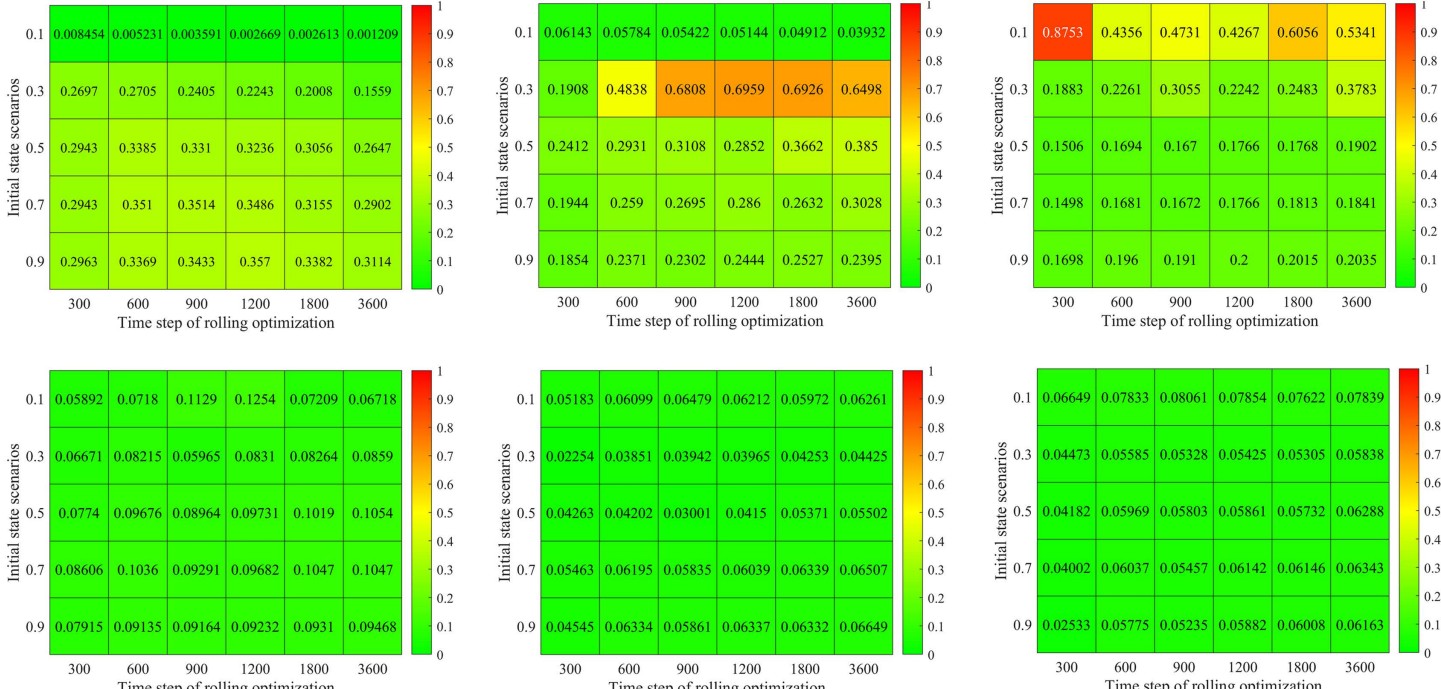

**Fig 16. Percentage of delay time reduced by rolling optimization strategy compared with the Synchro software. (a)** Demand scenarios 0.5. **(b)** Demand scenarios 1.0. **(c)** Demand scenarios 1.5. **(d)** Demand scenarios 2.0 **(e)** Demand scenarios 2.5. **(f)** Demand scenarios 3.0.

moderate-sized time step for rolling optimization, it's about 600s~1800s. (3) When the traffic demand (demand scenarios is 2.0~3.0) is high and initial state scenarios (0.7~0.9) are both high. the optimal average delay time in the given time period [17:00, 18:00) declines down with the growing time step of the rolling optimization. The optimal step for rolling optimization is 3600s.

After further investigating, it is found that decreasing the time step of rolling optimization has two opposing influences on the optimal average delay time in the given time period. When the rolling time step is smaller, it can generate lower average delay time because of the rapid response of the traffic signal control scheme to the time-varying demand. On the other hand, this benefit can be offset by the fact that smaller rolling time step will result in more local optimality which will affect the global optimality of average delay time. This is an important factor for the operator to consider when introducing the one-way carsharing system to a city. The existing traffic congestion will affect the system performance and design.

The reason is that the traffic congest at initial moment cannot dissipate immediately. The traffic congest may last for several time step of rolling optimization. Therefore, the traffic signal control scheme obtained in single optimization step time is not the global optimal traffic signal control scheme, which is leads to an increase in vehicle's average delay time. Meanwhile, if the time step of rolling optimization is large, the response of the traffic signal control scheme to the time-varying demand is weakened. Which is leads to an increase of vehicle's average delay time in some traffic signal cycle.

Fig 16 showed the percentage of delay time reduced by rolling optimization strategy compared with the Synchro software in different combinations of demand scenarios and initial states. It showed that the proposed model could also reduce the average delay time effectively. The average delay time can be reduced by 87.53% when the demand scenarios is 1.5 and the initial states is $0.1 \times C$.

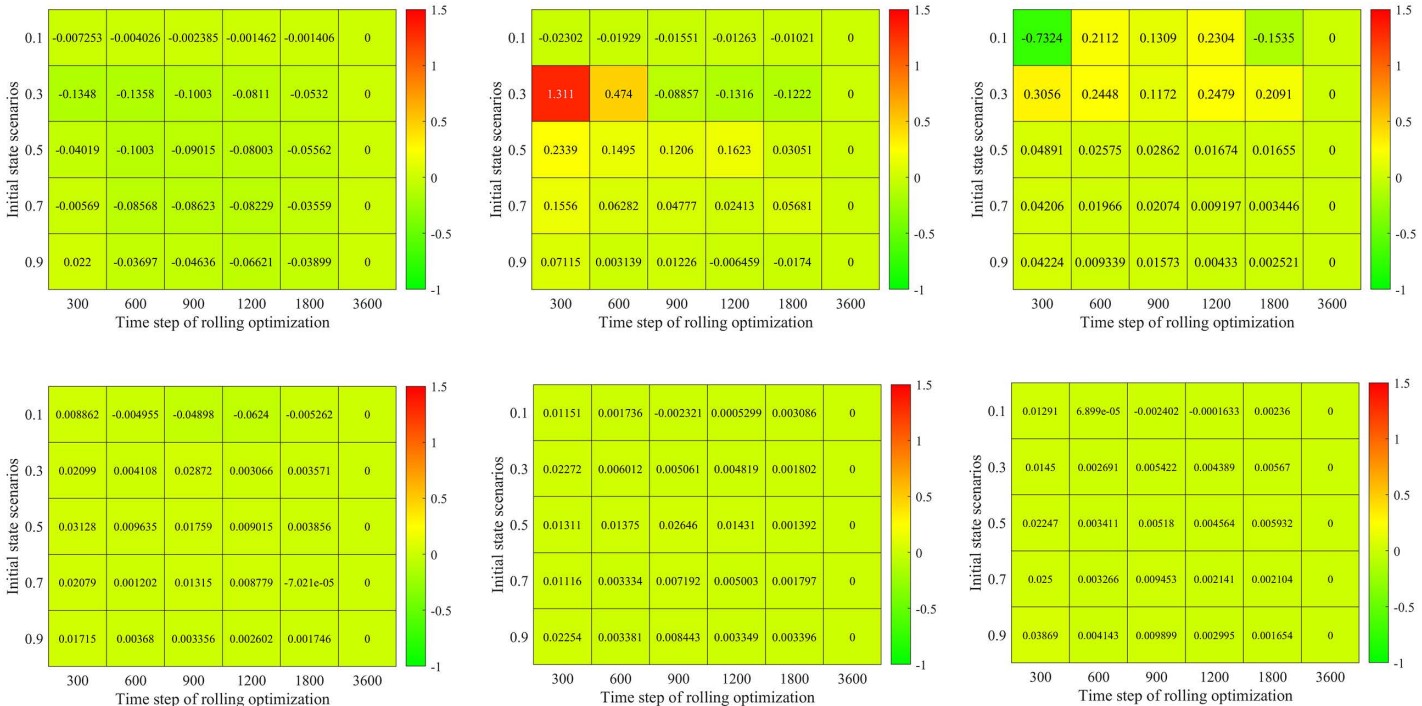

**Fig 17. Percentage of delay time reduced by rolling optimization strategy compared with no rolling optimization strategy. (a)** Demand scenarios 0.5. **(b)** Demand scenarios 1.0. **(c)** Demand scenarios 1.5. **(d)** Demand scenarios 2.0. **(e)** Demand scenarios 2.5. **(f)** Demand scenarios 3.0.

There are also some interesting findings between the proposed models with and without the rolling optimization strategy.

(1) As shown in Fig 15, There exists a optimal time step for rolling optimization to minimize the average delay time. The intersection needs to balance the dynamic adaptability of time-varying demand and the overall optimization during the traffic signal control strategy. (2) Under the same time step of rolling optimization, the impact of demand scenario and initial state on the optimization results is the same between the proposed models with and without the rolling optimization strategy.

Fig 17 shows the percentage of optimal vehicle's average delay time reduced by the proposed model with rolling optimization strategy compared with the proposed model without rolling optimization strategy in different combinations of demand and initial states. It shows that: (1) under certain traffic conditions, the optimization method with rolling strategy can further improve the optimal vehicle's average delay time. For example, the reduction of vehicle's average delay time is 73.24% when the time step of rolling optimization is 300s at the low traffic intensity (the initial states is $0.1 \times C$, the demand scenarios is 1.5). (2) When the traffic demand (demand scenarios is $0.5 \sim 1.0$) and initial state scenarios (0.1) are both slightly, the optimal average delay time in the given time period [17:00, 18:00) climbs up with the growing time step of the rolling optimization.(3) When the traffic demand (demand scenarios is $1.5 \sim 3.0$) is high and initial state scenarios ($0.5 \sim 0.9$) are both high, the proposed model with rolling optimization strategy can't further improve the optimal vehicle's average delay time.

## Extended optimization model analysis

In this section, we use the extended optimization model to obtain the signal timing scheme by considering the total cost. We briefly introduced the optimization results to show the feasibility of the model. The unit costs of delay and unit costs of

fuel consumption are set as $w_1 = 0.351$ Yuan/min and $w_2 = 7.71$ Yuan/liter, respectively. We use the rolling optimization model strategy for demand scenario 1 with different time step of rolling optimization.

Fig 18 a)-b) showed the total cost and reduction of total cost by our rolling optimization. Fig 18c)- d) and Fig 18e)- f) explained the vehicle's average delay time and average fuel consumption by our rolling optimization, respectively. From the results, we can find: (1) Compared with the Synchro software, the total cost of the proposed model with rolling optimization strategy can reduce by 65.56% at most when then the initial state scenario is $0.3 \times C$ and the step of rolling optimization is 1800s. (2) The proposed model with rolling optimization strategy can reduce the average delay time and fuel consumption simultaneously. Compared with the Synchro software, the proposed model's average delay time and fuel consumption can reduce by 69.45% and 62.40% at most when then the initial state scenario is $0.3 \times C$ and the step of rolling optimization is 1200s-1800s, respectively.

## Conclusion and Future Work

### Conclusion

The operates conditions in each direction of the intersection are the basis for developing the traffic control strategy. In this paper, we take the time-varying demand, state-dependent serviceability, and the randomness of arrival and service into account, established the feedback fluid flow queueing network to capture the dynamic performance in each direction intersection. Compared with the mean result of the 200 simulations, the average absolute error is 0.5152 vehicles, and the average relative error is 6.43%. That means the proposed model can capture the dynamic performance actually under different demand scenarios.

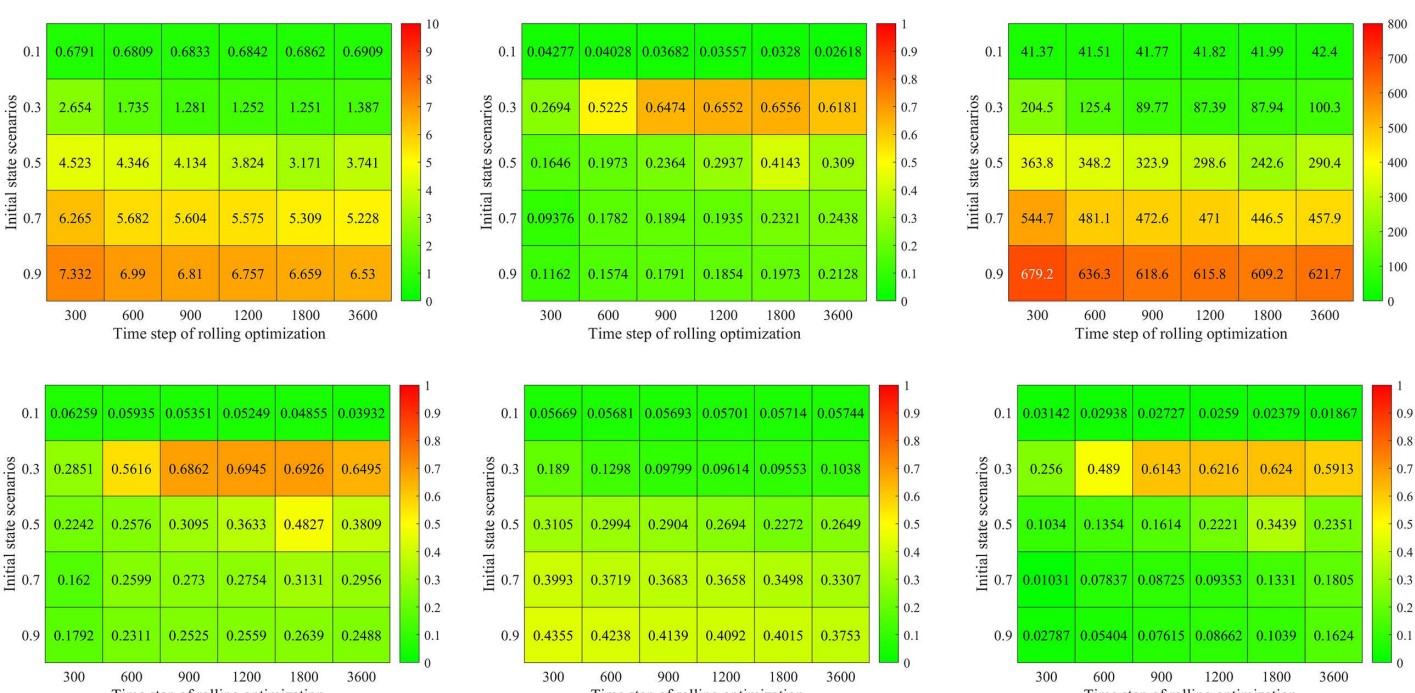

**Fig 18. The results of the extended optimization model. (a)** Total cost by the proposed model (Yuan/car). **(b)** Percentage of total cost reduced by proposed method. **(c)** Average delay time by proposed model (s/car). **(d)** Percentage of average delay time reduced by proposed method. **(e)** Average fuel consumption by proposed model (L/car). **(f)** Percentage of fuel consumption reduced by proposed method.

We also take the average delay time of vehicles as the objective function to obtain a better traffic control plan by the proposed basic model and the proposed model with a rolling optimization strategy, respectively. The results show that all of the proposed models can reduce the average delay time significantly. Compared with the Synchro software, the proposed basic model's average delay time can be reduced by 64.98%. And the average delay time obtained by the proposed model with a rolling optimization strategy can be reduced to achieve 87.53%.

The proposed model with a rolling optimization strategy can also minimize the total cost of average delay time and average fuel consumption. The results show that the average delay time and average fuel consumption are reduced simultaneously. Compared with the Synchro software, the proposed extension model's average delay time and fuel consumption can reduce by 69.45% and 62.40% at most under given traffic conditions, respectively. And the total cost of the proposed model can reduce by 65.56% at most.

After the numerical experiments, the following conclusions were reached: (1) The vehicle's average delay time is significantly affected by the initial states and demand scenarios. With the increases of initial states, the average delay time of vehicles increases significantly. Moreover, with the increase in demand scenarios, the average delay time of vehicles also increases significantly. (2) With the increase of the initial states, the reduction of average delay time shows a trend of increasing first and then decreasing at the lower demand scenarios. While at a higher demand scenario, the reduction of average delay time decreases with the increase of initial states. When the traffic demand is minimal, the average delay time of vehicles is small, so there is little room for average delay time improvement. However, with the increase of traffic intensity, adjusting the signal control plan according to the needs of different directions can better use the intersection's time and space resources, so the vehicle's average delay time is significantly improved. Furthermore, when the traffic intensity is high, the intersection's time and space resources have been fully utilized. It is difficult to reduce the delay time of vehicles by optimizing the signal control scheme. (3) Compare with Synchro software, the green time of each phase shows the same trend, first increasing and then decreasing as the increase of the initial states. And the change of green time is also affected by the traffic demand. For example, the change of phase 1 is more than other phases because the traffic intensity is more enormous than the other phase. Moreover, the change of the green time is decreased as the demand scenarios increase. When the traffic intensity is high, the intersection's time and space resources have been fully utilized, and it is difficult to reduce the delay time of vehicles by optimizing the signal control scheme. (4) The results show the proposed model with a rolling optimization strategy can reduce the average delay time significantly. With the time step of the rolling optimization increase, the optimal average delay time shows a decreasing trend and then increasing. This shows that it is not that the smaller the time step of rolling optimization, the smaller the average delay time. Instead, there is an optimal time step of rolling optimization to minimize the average delay time.

## Future Work

The proposed fluid flow feedback queueing network model can capture the dynamic performance actually under different demand scenarios. Meanwhile, the PSFFA has higher computation efficiency. The traffic control strategy for large-scale congested urban road networks will be studied in future work based on computational efficiency.

Meanwhile, traffic control strategies under the proposed model are extensible to mixed traffic environments comprising both human-driven vehicles (HVs) and automated vehicles (AVs). Due to distinct driving behaviors (characterized by car-following models) between HVs and automated vehicles AVs, their traffic flow characteristics exhibit significant differences. To model mixed traffic environments with coexisting HVs and AVs, we propose implementing a multi-class queuing system. Within this framework: (1) Vehicle-type heterogeneity is accommodated through differentiated service characteristics. (2) Driving behavior disparities are captured via class-specific service parameters. Furthermore, the service rate in queuing systems depends on the average speed of vehicles within the system. Consequently, we will: (1) Investigate fundamental diagrams under mixed traffic conditions. (2) Calibrate average travel speeds for queuing model parameterization. Subsequently, by treating distinct vehicle types as multiple customer classes in the feedback queuing

network model, a multi-class queuing network model for signalized intersections will be developed to represent mixed traffic dynamics. Developing and refining multi-class queuing theory to characterize mixed traffic flow constitutes a primary research direction for our future work.

It should be noted that the proposed model still exhibits several limitations: (1)Inflow/outflow dynamics: The current formulation inadequately captures nuanced traffic flow interactions between adjacent road segments, requiring further refinement in modeling lane-specific inflow-outflow patterns.(2) Poisson arrival assumption: The premise of Poisson-distributed arrivals may not hold under signal-controlled conditions where vehicle platooning effects dominate, necessitating future relaxation of this constraint. (3) Scope restriction: The exclusive focus on signalized intersections leaves unsignalized intersections (e.g., roundabouts, stop-controlled junctions) unmodeled. Consequently, large-scale real-world applications demand extensions to characterize uncontrolled intersection dynamics.

As rightly pointed out by the reviewer, the assumption of Poisson arrivals employed in this paper may deviate from actual traffic conditions. Deviations may particularly occur during peak hours or under the influence of traffic signals, making it difficult to guarantee that traffic flow arrivals strictly adhere to a Poisson distribution. The paper assumed that vehicle arrivals follow a Poisson distribution, primarily following established practices in related works such as [20,46,47,49,60,75]. While we fully acknowledge the limitations of this assumption, it remains fundamental to our current modeling approach and cannot be fundamentally altered within the scope of this study in the short term. In subsequent research, we plan to develop a queueing network model for intersections based on the Phase-type (PH) distribution. This is motivated by the theoretical capability of the PH distribution to approximate arbitrary arrival patterns, offering superior applicability. Furthermore, the PH distribution is closely related to the exponential distribution; indeed, it serves as a generalization and extension of the exponential distribution. Therefore, developing an intersection queueing network model based on the PH distribution appears highly feasible.

## Author contributions

**Data curation:** Shengyang Jiao, Denghui Yang.

**Investigation:** Shengyang Jiao, Denghui Yang.

**Methodology:** Bin Zhao.

**Project administration:** Bin Zhao.

**Validation:** Yanni Ju.

**Visualization:** Yanni Ju.

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
