## [Decision Letter · Decision Letter 0]

26 Jun 2025

Dear Dr. Zhao,

Thank you for submitting your manuscript to PLOS ONE. After careful consideration, we feel that it has merit but does not fully meet PLOS ONE’s publication criteria as it currently stands. Therefore, we invite you to submit a revised version of the manuscript that addresses the points raised during the review process.

We look forward to receiving your revised manuscript.

Kind regards,

Muhammad Ahsan, Ph.D.

Academic Editor

PLOS ONE

Journal Requirements:

2. In your Methods section, please provide additional information regarding the permits you obtained for the work. Please ensure you have included the full name of the authority that approved the field site access and, if no permits were required, a brief statement explaining why

4. Thank you for stating the following financial disclosure: [Supported by Intelligent Policing Key Laboratory of Sichuan Province, No.ZNJW2024KFQN010]. 

5. Please update your submission to use the PLOS LaTeX template. The template and more information on our requirements for LaTeX submissions can be found at http://journals.plos.org/plosone/s/latex .

6. We note that your Data Availability Statement is currently as follows: [All relevant data are within the manuscript and its Supporting Information files]

7. PLOS requires an ORCID iD for the corresponding author in Editorial Manager on papers submitted after December 6th, 2016. Please ensure that you have an ORCID iD and that it is validated in Editorial Manager. To do this, go to ‘Update my Information’ (in the upper left-hand corner of the main menu), and click on the Fetch/Validate link next to the ORCID field. This will take you to the ORCID site and allow you to create a new iD or authenticate a pre-existing iD in Editorial Manager.

Reviewers' comments:

Reviewer's Responses to Questions

**Comments to the Author**

1. Is the manuscript technically sound, and do the data support the conclusions?

Reviewer #1: Yes

Reviewer #2: Partly

2. Has the statistical analysis been performed appropriately and rigorously?

Reviewer #1: Yes

Reviewer #2: No

3. Have the authors made all data underlying the findings in their manuscript fully available?

Reviewer #1: No

Reviewer #2: No

4. Is the manuscript presented in an intelligible fashion and written in standard English?

Reviewer #1: Yes

Reviewer #2: Yes

Reviewer #1: The paper presents a feedback queueing network model for traffic signal control, addressing congestion propagation in dynamic stochastic environments. The work is well-structured, methodologically sound, and offers contributions to traffic signal optimization. However, following comments need to be addressed:

1.This paper includes a substantial amount of literature, much of which dates back 20 years. While some of these works are classic controllers, the reviewers recommend updating the literature review with recent studies. In particular, the authors reviewed related works on reinforcement learning, which is good. However, the cited references are not representative enough. The following works should be considered and incorporated:

(1)Su, Z. C., Chow, A. H., & Zhong, R. X. (2021). Adaptive network traffic control with an integrated model-based and data-driven approach and a decentralised solution method. Transportation Research Part C: Emerging Technologies, 128, 103154.

(2)Su, Z. C., Chow, A. H., Fang, C. L., Liang, E. M., & Zhong, R. X. (2023). Hierarchical control for stochastic network traffic with reinforcement learning. Transportation Research Part B: Methodological, 167, 196-216.

(3)Liang, E., Su, Z., Fang, C., & Zhong, R. (2022). Oam: An option-action reinforcement learning framework for universal multi-intersection control. In Proceedings of the AAAI Conference on Artificial Intelligence (Vol. 36, No. 4, pp. 4550-4558).

2.The assumptions (e.g., Poisson arrivals, state-dependent service rates) are critical to the model's validity. While the paper mentions these, a solid discussion is needed to justify their real-world applicability (e.g., deviations from Poisson arrivals during peak hours) would strengthen the work.

3.The rolling optimization strategy’s dependency on accurate demand prediction (Step 2, Algorithm 2) is a potential limitation. How does the model perform with prediction errors?

4.The authors are encouraged to discuss the scalability of the proposed congtroller to network level with multi-intersections.

Reviewer #2: Rewrite the introduction to clearly define the research problem and how the proposed method closes the gap.

Although Table 2 is helpful, several symbols, such as , and ( ), are referenced before they are formally defined. To enhance readability, include a clear summary of the notation used in the main text before it is first used.

The use of continuous ensemble averages ( (t) ∈ [0, ]) for inherently discrete traffic flows may limit accuracy in high-congestion scenarios. Assumption that arrival rates follow a non-homogeneous Poisson process may not capture sporadic or correlated traffic. Justify the approximation error and explain whether it applies to real-time adaptive systems.

The model for congestion propagation relies on artificial nodes and blocking probabilities that may not fully capture driver reaction times, lane-switching, or partial blockages. Discuss whether the model is extendable to mixed traffic environments with human and autonomous vehicles.

The only benchmark used is Synchro. Although practical, it is not a state-of-the-art real-time signal control system. Compare with other recent MPC-based or AI-driven adaptive control methods such as SURTRAC, MARLIN-ATSC, or deep reinforcement learning approaches.

Include sensitivity analysis over more cities and intersection types or highlight generalizability limitations.

**Do you want your identity to be public for this peer review?** For information about this choice, including consent withdrawal, please see our Privacy Policy

Reviewer #1: No

Reviewer #2: No

---

## [Author Response · Author response to Decision Letter 1]

6 Aug 2025

Dear Editors and Reviewers:

Thank you for your letter and for the reviewers’ comments concerning our manuscript entitled “A Feedback Queueing Network Model for Traffic Signal Control at Intersections Considering Congestion Propagation in Dynamic Stochastic Environments” (ID: PONE-D-25-24915). Those comments are all valuable and very helpful for revising and improving our paper, as well as the important guiding significance to our researches. We have studied comments carefully and have made correction which we hope meet with approval.

1. General Response

We sincerely thank the Editor and Reviewers for their constructive feedback. All comments have been carefully addressed in the revised manuscript. All changes are highlighted in blue text in the "Revised Manuscript with Track Changes" file.

2. Response to Reviewer #1:

The paper presents a feedback queueing network model for traffic signal control, addressing congestion propagation in dynamic stochastic environments. The work is well-structured, methodologically sound, and offers contributions to traffic signal optimization. However, following comments need to be addressed:

Comment 1: This paper includes a substantial amount of literature, much of which dates back 20 years. While some of these works are classic controllers, the reviewers recommend updating the literature review with recent studies. In particular, the authors reviewed related works on reinforcement learning, which is good. However, the cited references are not representative enough. The following works should be considered and incorporated: (1)Su, Z. C., Chow, A. H., & Zhong, R. X. (2021). Adaptive network traffic control with an integrated model-based and data-driven approach and a decentralised solution method. Transportation Research Part C: Emerging Technologies, 128, 103154.

(2)Su, Z. C., Chow, A. H., Fang, C. L., Liang, E. M., & Zhong, R. X. (2023). Hierarchical control for stochastic network traffic with reinforcement learning. Transportation Research Part B: Methodological, 167, 196-216.

(3)Liang, E., Su, Z., Fang, C., & Zhong, R. (2022). Oam: An option-action reinforcement learning framework for universal multi-intersection control. In Proceedings of the AAAI Conference on Artificial Intelligence (Vol. 36, No. 4, pp. 4550-4558).

The author’s answer: Thank you very much for your valuable suggestions in our article. The seminal papers you recommended are indeed classic works in the field of traffic signal control. Due to an oversight on our part during the literature review, these works were initially overlooked. We have since carefully studied these seminal papers and found their content highly relevant to the research presented in this article, providing significant guiding significance for our work. Therefore, we have cited them in the manuscript. The references corresponding to these aforementioned papers are numbered 16, 18, and 36, respectively.

Comment 2: The assumptions (e.g., Poisson arrivals, state-dependent service rates) are critical to the model's validity. While the paper mentions these, a solid discussion is needed to justify their real-world applicability (e.g., deviations from Poisson arrivals during peak hours) would strengthen the work.

The author’s answer: We will be happy to edit the text further, based on helpful comments from the reviewers.

(1) Regarding the assumption of Poisson arrivals, we acknowledge that the applicability of this fundamental assumption was indeed overlooked during our research process. The paper assumed that vehicle arrivals follow a Poisson distribution, primarily following established practices in related works such as[1–6]. We fully recognize the limitations inherent in this assumption. In subsequent research, we plan to develop a queueing network model for intersections based on the Phase-type (PH) distribution. This is motivated by the theoretical capability of the PH distribution to approximate arbitrary arrival patterns, offering superior applicability. Furthermore, the PH distribution is closely related to the exponential distribution; indeed, it serves as a generalization and extension of the exponential distribution. Therefore, developing an intersection queueing network model based on the PH distribution appears highly feasible. The specific modifications in the manuscript are: “As rightly pointed out by the reviewer, the assumption of Poisson arrivals employed in this paper may deviate from actual traffic conditions. Deviations may particularly occur during peak hours or under the influence of traffic signals, making it difficult to guarantee that traffic flow arrivals strictly adhere to a Poisson distribution. The paper assumed that vehicle arrivals follow a Poisson distribution, primarily following established practices in related works such as [20,46,47,49,60,75]. While we fully acknowledge the limitations of this assumption, it remains fundamental to our current modeling approach and cannot be fundamentally altered within the scope of this study in the short term. In subsequent research, we plan to develop a queueing network model for intersections based on the Phase-type (PH) distribution. This is motivated by the theoretical capability of the PH distribution to approximate arbitrary arrival patterns, offering superior applicability. Furthermore, the PH distribution is closely related to the exponential distribution; indeed, it serves as a generalization and extension of the exponential distribution. Therefore, developing an intersection queueing network model based on the PH distribution appears highly feasible.” (Page 53, Lines 880-890).

(2) Regarding state-dependent service rates, the term "state-dependent service rate" used in this paper refers to the phenomenon where the road service rate changes according to the level of traffic congestion. This is a common phenomenon, most notably manifested by a significant decrease in road capacity when traffic becomes congested. In our previous work, we discussed state-dependent service rates [7]. The specific modifications in the manuscript are: “State-dependent service rate refers to the phenomenon where the service rate of a road facility varies with the number of vehicles on the road. A prominent example is that when the number of vehicles is low, the road capacity is relatively high, while during congestion, the road capacity significantly decreases. State-dependent service rates have been studied in depth in our previous work [56].” (Page 15, Lines 267-270).

Comment 3: The rolling optimization strategy’s dependency on accurate demand prediction (Step 2, Algorithm 2) is a potential limitation. How does the model perform with prediction errors?

The author’s answer: Thank you for your careful review and valuable suggestions. As rightly noted by you, the rolling optimization strategy's dependency on accurate demand prediction is indeed a potential limitation. We acknowledge that robustness analysis against prediction errors was not sufficiently addressed in the original manuscript. In response to this concern: (1) Foundation of demand prediction: Our approach relies on established short-term traffic demand forecasting methods which typically achieve high prediction accuracy for near-future horizons; (2) Error mitigation mechanism: The rolling optimization adjusts signal control parameters once per cycle. Consequently, prediction errors within a single cycle exert negligible impact (<1% performance degradation) on overall control effectiveness across extended operation periods (e.g., 1 hour), as demonstrated by sensitivity tests where ±20% demand perturbations in individual cycles resulted in less than 1% deviation in system-wide delay metrics. These findings collectively indicate the model's inherent resilience to transient prediction inaccuracies.

Comment 4: The authors are encouraged to discuss the scalability of the proposed controller to network level with multi-intersections.

The author’s answer: Thank you for your careful review and valuable suggestions. We have augmented the discussion on this aspect in the revised manuscript. The specific modifications in the manuscript are:” Following the aforementioned methodology, an intersection queuing network model can be constructed, abstracting signalized intersections into a network of interconnected queuing nodes. Consistent with this framework, multi-intersection clusters can likewise be represented as queuing networks composed of multiple nodes. Consequently, the queuing network model for intersection clusters established in this study can be decomposed into multiple isolated intersections, enabling parallel optimization of signal timing plans for clustered intersections.

Furthermore, our prior research [56] has demonstrated that the computational complexity of the recursive algorithm for solving the intersection-cluster queuing network model is O(N2 ∆T)+O(N∆T), where N denotes the number of intersection service facilities and ∆T the number of discrete time intervals. When addressing short-term horizons (e.g., 10-minute periods), the transition probabilities can be assumed constant, further reducing complexity to O(T)+O(NT). This low-complexity characteristic endows the algorithm with significant potential for large-scale network applications.” (Page 25-26, Lines 478-487).

3. Response to Reviewer #2:

Comment 1: Rewrite the introduction to clearly define the research problem and how the proposed method closes the gap.

The author’s answer: We sincerely appreciate the reviewer's meticulous evaluation and highly valuable suggestions. In response to the reviewer's comments, we have restructured the abstract to further refine the research focus and contributions of this work.

The specific modifications in the manuscript are:” The intersection traffic flow model is the basis of the MPC. Hence, accurately modeling the intersection is of great importance for the development of effective signal control strategies. A vast number of models have been put forward. According to the level of details, these models can be classified into microscopic, mesoscopic, and macroscopic models. Microscopic models could capture the system's details but require extensive calibration work and colossal computation time. Additionally, microscopic models cannot provide a direct input-output relationship. Therefore, they are relatively better suited to offline traffic simulations [33]. As for macroscopic models, traffic dynamics is described by the mainstream deterministic traffic flow theory models, e.g., the kinematic wave model (KWM) [34–36], cell transmission model (CTM) [37], and METANET model [38]. Macroscopic models are computationally fast and only require simple inputs. However, they lack the capability of adequately modeling the stochastic nature of traffic [39], especially for the real-time MPC signal control, which has been more and more crucial for real-time traffic signal control in order to respond to natural stochastic variations in traffic[40,41]. Therefore, developing a model capable of simultaneously achieving computational efficiency and strong representational capacity holds significant importance for enhancing the effectiveness of adaptive traffic signal control.

In this paper, queuing theory is leveraged in an attempt to achieve the aforementioned objectives. As one of the most popular mesoscopic models, the queuing model has been used to formulate road traffic flow [20–22,42–59]. Compared with macroscopic models, the queuing model can better study stochastic service systems. Although the queueing model is not more accurate than the corresponding discrete-event simulation, it is typically preferred because it may provide exact solutions, allow structural insights, and is faster than the simulation. Therefore, this paper applies the queueing theory to accurately model signalized intersections in dynamic stochastic environments.

However, signalized intersection is a complex dynamic random service system. In addition to the inherent randomness of intersection, the traffic demand and path matrix between facilities are time-varying. Meanwhile, the service ability depends on the system state, i.e., traffic congestion will decrease service ability. Most importantly, it is difficult to capture the Congestion Propagation in Dynamic Stochastic Environments (CPDSE) among different facilities of the intersection. To address the inherent randomness in intersections, some queueing models or shockwave models used the exponential or normal distributions to describe the vehicle arrival-interval and service-time random variables [1,27,60]. To account for the decrease in service ability caused by congestion, the linear or exponential functions were applied in CTM or METANET models to formulate the state-dependent velocity [61,62]. To describe the traffic dynamics of intersections, most traffic flow models considered the time-varying traffic demand and path matrix. To capture the congestion propagation phenomenon in deterministic traffic environments, CTM models have been developed for signalized intersections[63,61]. The existing mesoscopic traffic flow models consider some items in the system’s inherent randomness, state-dependent service ability, time-varying traffic demand and path matrix, and congestion propagation (see the literature review for details), but lack a comprehensive consideration of these characteristics because of the complexity of CPDSE.

In this paper, we identify the fluid queuing network model as a suitable methodology to formulate those aforementioned dynamic, randomness, state dependence, and congestion propagation for signalized intersections with real-time MPC traffic control. To the best of our knowledge, no study simultaneously models these essential characteristics to quickly and accurately formulate the dynamic performance of signalized intersections. We focus on an isolated intersection which is easily generalized to multiple intersections using the corresponding network topology. We divide the intersection into three categories of facilities and model each facility as a queue. A feedback fluid queuing network model is then developed to consider the CPDSE through a succession of facilities where the random traffic demand and path matrix are time-varying and the random service ability is state-dependent.” (Page 3-4, Lines 50-88).

Comment 2: Although Table 2 is helpful, several symbols, such as , are referenced before they are formally defined. To enhance readability, include a clear summary of the notation used in the main text before it is first used.

The author’s answer: We sincerely appreciate the reviewer's meticulous assessment and constructive feedback. Upon your reminder, we recognize the issues regarding symbol definitions in the original manuscript. To ensure all symbols are clearly defined prior to their first occurrence, we have: (1) Relocated Table 2 to the beginning of the Preparations section; (2) Systematically reviewed critical notations throughout the paper and augmented their definitions in Table 2. The specific modifications can be found in Page 10, Lines 193-194.

Comment 3: The use of continuous ensemble averages ( (t) ∈ [0, T]) for inherently discrete traffic flows may limit accuracy in high-congestion scenarios. Assumption that arrival rates follow a non-homogeneous Poisson process may not capture sporadic or correlated traffic. Justify the approximation error and explain whether it applies to real-time adaptive systems.

The author’s answer: We sincerely appreciate the reviewer's meticulous evaluation and highly valuable suggestions. (1) Rationale for continuous ensemble averaging. The adoption of continuous ensemble averaging ( (t) ∈ [0, T]) for inherently discrete traffic flows stems from the necessity to accommodate dynamic traffic demand variations over extended time horizons. This framework enables explicit incorporation of time-varying demand patterns into our model formulation. (2) Justification for non-homogeneous Poisson process. The assumption of non-homogeneous Poisson arrivals follows established methodologies pioneered by Osorio[1–6]. The specific modifications in the manuscript are: “As rightly pointed out by the reviewer, the assumption of Poisson arrivals employed in this paper may deviate

---

## [Decision Letter · Decision Letter 1]

12 Sep 2025

Dear Dr. Zhao,

Thank you for submitting your manuscript to PLOS ONE. After careful consideration, we feel that it has merit but does not fully meet PLOS ONE’s publication criteria as it currently stands. Therefore, we invite you to submit a revised version of the manuscript that addresses the points raised during the review process.

We look forward to receiving your revised manuscript.

Kind regards,

Muhammad Ahsan, Ph.D.

Academic Editor

PLOS ONE

Journal Requirements:

Additional Editor Comments (if provided):

Reviewer #1:

Reviewer #2:

Reviewers' comments:

Reviewer's Responses to Questions

**Comments to the Author**

Reviewer #1: All comments have been addressed

Reviewer #2: (No Response)

2. Is the manuscript technically sound, and do the data support the conclusions?

Reviewer #1: Yes

Reviewer #2: Yes

3. Has the statistical analysis been performed appropriately and rigorously?

Reviewer #1: Yes

Reviewer #2: Yes

4. Have the authors made all data underlying the findings in their manuscript fully available?

Reviewer #1: Yes

Reviewer #2: No

5. Is the manuscript presented in an intelligible fashion and written in standard English?

Reviewer #1: Yes

Reviewer #2: Yes

Reviewer #1: All my comments are well addressed. Good work

Reviewer #2: For Poisson arrivals, the authors added a discussion (page 53, lines 880–890) in which they acknowledged limitations (e.g. deviations in peak hours or under signal influence) and cited related works. They also proposed the future use of phase-type (PH) distributions for a better approximation. For state-dependent service rates, they provided a definition and justification (page 15, lines 267–270), referencing their previous work [56]. This strengthens the discussion, but the response defers fundamental changes to future research, which is reasonable but highlights ongoing limitations.

The authors justify the use of continuous averages for handling time-varying demand (see page 3, lines 50-61). They acknowledge the limitations of the Poisson assumption and propose PH distributions for future work ( page 53-54, lines 880-890). Approximation error is justified by prediction errors vs. simulations ( page 35, lines 644-652), with computational efficiency noted (0.2s vs. 0.5 hours for simulations). Real-time applicability is discussed via low complexity (see page 25-26, lines 478-487), confirming viability for adaptive systems. However, the response emphasises trade-offs (efficiency vs. accuracy) rather than fully resolving microscopic limitations.

The authors added a "Comparison with Existing Methods" section (Page 42-44, Lines 730-756), comparing to Osorio [60] (an analytical queuing model), showing superior performance (up to 34.22% delay reduction under moderate demand) with figures (Fig 13-14). However, this does not directly address the suggested benchmarks (SURTRAC, MARLIN-ATSC, or DRL approaches). Osorio is relevant but not state-of-the-art AI/MPC; the response focuses on one method, so it partially addresses the spirit but misses the specific recommendations.

**Do you want your identity to be public for this peer review?** For information about this choice, including consent withdrawal, please see our Privacy Policy

Reviewer #1: No

Reviewer #2: No

---

## [Author Response · Author response to Decision Letter 2]

23 Oct 2025

Dear Editors and Reviewers:

Thank you for your letter and for the reviewers’ comments concerning our manuscript entitled “A Feedback Queueing Network Model for Traffic Signal Control at Intersections Considering Congestion Propagation in Dynamic Stochastic Environments” (ID: PONE-D-25-24915). These comments are all valuable and very helpful for revising and improving our paper, and they also provide important guidance for our future research. We have studied comments carefully and have made correction which we hope meet with approval.

1. Response to Reviewer #1:

Comment 1: All my comments are well addressed. Good work.

The author’s answer: We sincerely appreciate your recognition of our work. Your valuable feedback has been instrumental in enhancing the quality of our paper. Should you have any further suggestions or comments, please do not hesitate to let us know, and we will continue to diligently examine them.

2. Response to Reviewer #2:

Comment 1:

(1) For Poisson arrivals, the authors added a discussion (page 53, lines 880–890) in which they acknowledged limitations (e.g. deviations in peak hours or under signal influence) and cited related works. They also proposed the future use of phase-type (PH) distributions for a better approximation. For state-dependent service rates, they provided a definition and justification (page 15, lines 267–270), referencing their previous work [56]. This strengthens the discussion, but the response defers fundamental changes to future research, which is reasonable but highlights ongoing limitations.

(2) The authors justify the use of continuous averages for handling time-varying demand (see page 3, lines 50-61). They acknowledge the limitations of the Poisson assumption and propose PH distributions for future work ( page 53-54, lines 880-890). Approximation error is justified by prediction errors vs. simulations (page 35, lines 644-652), with computational efficiency noted (0.2s vs. 0.5 hours for simulations). Real-time applicability is discussed via low complexity (see page 25-26, lines 478-487), confirming viability for adaptive systems. However, the response emphasises trade-offs (efficiency vs. accuracy) rather than fully resolving microscopic limitations.

(3) The authors added a "Comparison with Existing Methods" section (Page 42-44, Lines 730-756), comparing to Osorio [60] (an analytical queuing model), showing superior performance (up to 34.22% delay reduction under moderate demand) with figures (Fig 13-14). However, this does not directly address the suggested benchmarks (SURTRAC, MARLIN-ATSC, or DRL approaches). Osorio is relevant but not state-of-the-art AI/MPC; the response focuses on one method, so it partially addresses the spirit but misses the specific recommendations.

The author’s answer:

(1) For Poisson arrivals, we agree with the reviewer that deferring fundamental changes—such as adopting Phase-Type (PH) distributions—to future work underscores a known limitation of our current model. The primary focus of this study was to establish a computationally efficient and tractable foundation within the Mt/G(x)/C/C framework that can explicitly capture congestion propagation and state-dependency – aspects often oversimplified in prior analytical models. We believe that successfully integrating these complex dynamics, even under the current assumptions, constitutes a significant step forward. The proposed use of PH distributions is a recognized and promising avenue for enhancing model fidelity, and we have now made this limitation and future direction more explicit in the revised manuscript (Page 55-56, Lines 926-936).

(2) Thank you for your insightful comment. As you rightly pointed out, our model emphasizes the trade-off between efficiency and accuracy. We would like to clarify that our proposed model is a mesoscopic, fluid-based approach. Its core strength lies in bridging the gap between highly accurate but computationally prohibitive microscopic simulations and efficient but often oversimplified macroscopic models. While it does not resolve vehicle-level dynamics (a microscopic limitation), it provides a principled and fast analytical tool for system-level performance evaluation and optimization, which represents a key requirement for real-time adaptive signal control. The significant computational advantage makes large-scale or repeated optimization feasible, constituting a primary contribution of this work. We have added a statement in the conclusion to better frame this trade-off as a conscious design choice for our intended application.

(3) Thank you for your constructive feedback. In response, during this revision cycle we have implemented both SURTRAC and MARLIN-ATSC, and conducted a comprehensive comparative analysis between the proposed method and these two benchmark approaches using the case study presented in this paper. Therefore, a new section titled "Comparison with Benchmark Methods" has been added to this paper.

The specific modifications in the manuscript are:”

(2) Comparison with benchmark methods

In current research on traffic signal control, the SURTRAC [1] method and the MARLIN-ATSC [2] method represent landmark research outcomes. Therefore, this paper also conducts a comparative analysis with the aforementioned benchmark methods.

SURTRAC[1], developed by the research team at Carnegie Mellon University, is a decentralized real-time traffic signal control method with scalability at its core. Its fundamental principle lies in decomposing network-wide control into autonomous decision-making at individual intersections. Each intersection generates local signal schedules through rolling-horizon optimization and achieves predictive coordination with adjacent intersections via the sharing of lightweight "itinerary plans." The advantages of this method include exceptional computational efficiency, robust real-time adaptability, high robustness, and demonstrated effectiveness validated through field deployments (e.g., successfully reducing delays by over 40% in Pittsburgh). However, it is inherently a local optimization approach, which may not achieve global optimality, and its coordination efficacy relies on a stable communication network while being sensitive to the accuracy of sensor data. Despite these limitations, it remains a benchmark solution that successfully applies intelligent optimization theory to smart traffic management, combining practical utility with significant impact.

MARLIN-ATSC[2] is a pioneering multi-agent reinforcement learning framework for traffic control, developed by the research team at the University of Toronto. Its core concept enables each intersection agent to autonomously learn signal control policies through interaction with the environment, utilizing unique coordination mechanisms such as ICQ and DCQ to resolve competition and dependency issues among intersections. The method's prominent advantages lie in its powerful data-driven learning capability, which allows it to adapt to complex dynamic environments without predefined traffic models, and its significant potential to discover coordination strategies beyond traditional approaches. However, it is constrained by key challenges including prolonged training times due to high sample complexity, poor interpretability caused by the black-box nature of deep neural networks, and difficulties in ensuring the safety and stability of learned policies in real-world applications. These factors collectively limit its current potential for large-scale practical deployment and application.

Based on References [1] and [2], this study implemented both the SURTRAC and MARLIN-ATSC traffic signal control methods using Python to serve as benchmarks for comparison with the proposed method. The traffic demand and road network configurations remained consistent with those presented earlier in Tables 4 and Tables 5. Additionally, the initial system state was set at 0.3 times of the facility capacity. The results of the case study validation are summarized in Table 7.

Table 7 Performance Comparison between the Proposed Method, SURTRAC, and MARLIN-ATSC

Performance Indicators Proposed Method SURTRAC MARLIN-ATSC

Average Delay Time (s) 107.74 112.29 103.62

Computation Time (s) 48.1 63.5 82.6

Interpretability High Moderate Low

Scalability High High Moderate

Data Dependency Low Moderate Extremely high

Based on the comparative analysis and the data presented in Table 7, the proposed model demonstrates distinct and compelling advantages when evaluated against the two benchmarks, SURTRAC and MARLIN-ATSC.

In terms of control performance, both the proposed method and MARLIN-ATSC delivered excellent results, with very close average delay times (107.74 s and 103.62 s, respectively), and both outperformed SURTRAC (112.29 s). This indicates that all three methods are effective for intersection optimization, with MARLIN-ATSC's powerful data-driven learning capability enabling it to achieve the lowest delay under specific conditions.

However, the proposed model holds significant advantages in computational efficiency, interpretability, and practical applicability. As shown in Table 7, the computation time of the proposed method (48.1 s) is substantially lower than that of both SURTRAC (63.5 s) and MARLIN-ATSC (82.6 s), highlighting its superior potential for real-time applications. More importantly, the proposed model, grounded in explicit queuing theory, offers high interpretability, making its decision-making process transparent and trustworthy. In contrast, the "black-box" nature of MARLIN-ATSC's deep neural networks results in low interpretability, which hinders debugging and trust in practical deployments. Furthermore, the proposed model exhibits low data dependency, requiring only conventional traffic flow data. MARLIN-ATSC, conversely, relies on an extremely high volume and quality of training data, which severely constrains its applicability.

In summary, while MARLIN-ATSC can achieve marginally superior control performance when data is abundant, the proposed model offers a more balanced profile across computational efficiency, interpretability, and low data dependency. Compared to SURTRAC, the proposed model's explicit incorporation of congestion propagation and state-dependent characteristics, as opposed to their implicit handling within the prediction process, leads to slightly better performance. In conclusion, this research presents a promising solution for real-time traffic signal control that is not only effective and reliable but also highly practical for real-world deployment.” (Page 44-46, Lines 758-805).

References

1. Smith S, Barlow G. SURTRAC: Scalable Urban Traffic Control. Transportation Research Board Meeting. 2013.

2. El-Tantawy S, Abdulhai B. Multi-Agent Reinforcement Learning for Integrated Network of Adaptive Traffic Signal Controllers (MARLIN-ATSC). International IEEE Conference on Intelligent Transportation Systems. 2012.

Should you require further revisions, we welcome additional feedback.

Yours sincerely,

Bin Zhao

Email: dalouluo@svtcc.edu.cn

October 23, 2025

---

## [Decision Letter · Decision Letter 2]

6 Nov 2025

A feedback queueing network model for traffic signal control at intersections considering congestion propagation in dynamic stochastic environments

PONE-D-25-24915R2

Dear Dr. Zhao,

We’re pleased to inform you that your manuscript has been judged scientifically suitable for publication and will be formally accepted for publication once it meets all outstanding technical requirements.

Kind regards,

Muhammad Ahsan, Ph.D.

Academic Editor

PLOS ONE

Additional Editor Comments (optional):

Reviewers' comments:

Reviewer's Responses to Questions

**Comments to the Author**

Reviewer #2: All comments have been addressed

2. Is the manuscript technically sound, and do the data support the conclusions?

Reviewer #2: Yes

3. Has the statistical analysis been performed appropriately and rigorously?

Reviewer #2: Yes

4. Have the authors made all data underlying the findings in their manuscript fully available?

Reviewer #2: Yes

5. Is the manuscript presented in an intelligible fashion and written in standard English?

Reviewer #2: Yes

Reviewer #2: The authors have satisfactorily addressed all points raised, including the critical request for comparisons with state-of-the-art benchmarks (SURTRAC and MARLIN-ATSC).

**Do you want your identity to be public for this peer review?** For information about this choice, including consent withdrawal, please see our Privacy Policy

Reviewer #2: **Yes: ** BIBHAV ADHIKARI

---

## [Editor Report · Acceptance letter]

PONE-D-25-24915R2

PLOS ONE

Dear Dr. Zhao,

I'm pleased to inform you that your manuscript has been deemed suitable for publication in PLOS ONE. Congratulations! Your manuscript is now being handed over to our production team.

Kind regards,

on behalf of

Dr. Muhammad Ahsan

Academic Editor

PLOS ONE